# Characterization of morphological units in a small, forested stream using close-range remotely piloted aircraft imagery

Carina Helm[1], Marwan A. Hassan[1], and David Reid[1]

[1]Department of Geography, University of British Columbia, Vancouver, BC, Canada

**Correspondence:** Carina Helm (helm.carina@gmail.com)

**Abstract.**

Forested, gravel bed streams possess complex channel morphologies which are difficult to objectively characterise. The spatial scale necessary to adequately capture variability in these streams is often unclear, as channels are governed by irregularly spaced features and episodic processes. This issue is compounded by the high cost and time-consuming nature of field surveys in these complex fluvial environments. In larger streams, remotely piloted aircraft (RPAs) have proven to be effective tools for characterizing channels at high resolutions over large spatial extents, but to date their use in small, forested streams with closed forest canopies has been limited. This paper seeks to demonstrate an effective method for classifying channel morphological units in small, forested streams and for providing information on the spatial scale necessary to capture the dominant spatial morphological variability of these channels. This goal was achieved using easily extractable data from close-range RPA imagery collected under the forest canopy (flying height = 5 – 15 m above ground level) in a small (width = 10 – 15 m) stream along its 3 km of salmon-bearing channel. First, the accuracy and coverage of RPAs for extracting channel data were investigated through a sub-canopy survey. From this survey data, relevant cross-sectional variables (hydraulic radius, sediment texture and channel slope) were extracted from high resolution point clouds and DEMs of the channel, and used to characterise channel unit morphology using a principal component analysis-clustering (PCA-clustering) technique. Finally, the length scale required to capture dominant morphological variability was investigated from an analysis of morphological diversity along the channel. The results demonstrate that sub-canopy RPA surveys provide a viable alternative to traditional ground-based survey approaches for mapping morphological units, with 87% coverage of the main channel stream bed achieved. The PCA-clustering analysis provided a comparatively objective means of classifying channel unit morphology with a correct classification rate of 85%. An analysis of the morphological diversity along the surveyed channel indicates that reaches of at least 15 bankfull width equivalents are required to capture the channel's dominant morphological heterogeneity. Altogether, the results provide a precedent for using RPAs to characterise the morphology and diversity of forested streams under dense canopies.

## 1  Introduction

Channel morphological units such as pools and riffles constitute the building blocks of reach scale channel morphologies (Buffington and Montgomery, 2013), with spatial variability in these units providing critical habitat diversity. As a result, characterization of morphological units is a goal of many habitat-based classification schemes (e.g. Hawkins et al., 1993).

Morphological unit classification may be particularly important in forested, gravel bed streams, where episodic and transient geomorphological processes (Pryor et al., 2011; Wohl and Brian, 2015; Hassan et al., 2019), can lead to a high degree of channel complexity even within a relatively homogeneous channel type (Madej, 1999; Nelson et al., 2010; Gartner et al., 2015). Within these streams, classification schemes can serve an important role in facilitating discussions on stream management (Buffington and Montgomery, 2013). This is evident in the array of classification schemes proposed to characterise channel types and morphological units for both geomorphologists and ecologists alike (e.g. Hawkins et al., 1993; Rosgen, 1994; Montgomery and Buffington, 1997; Brierly and Fryirs, 2005). A common challenge of these classification approaches, however, is their descriptive nature (Buffington and Montgomery, 2013; Hassan et al., 2017) and that their implementation can be subjective, differing between classifiers.

Challenges in objectively classifying morphological units are further compounded by difficulties in determining the appropriate spatial extent for capturing the primary structural variability that influences geomorphological and ecological processes at the reach or basin scale. While approaches are often taken to select 'representative sites' when the characterisation of channel variables is necessary (Harrelson et al., 1994; Bisson et al., 2006), site selection is often based on a narrow subset of metrics (e.g. gradient, see Montgomery and Buffington, 1998) and 'rules of thumb' are frequently used to define the spatial extent of the surveyed area (Bisson et al., 2006). Furthermore, traditional survey techniques often limit classification to short, accessible channel areas due to time and cost constraints, and these limitations may bias our understanding of the larger river network as a result of missing important channel areas and processes (Fausch et al., 2002; Hugue et al., 2016). Given the logistical difficulty and cost of undertaking field surveys in small, forested, gravel-bed streams, a more precise approach for site selection and objective technique for classifying morphological units is warranted.

Traditionally, characterization and classification of channels through field surveys has required the use of a variety of GPS-based tools and linear-survey methods involving automatic levels, theodolites, and total-stations (e.g. Bangen et al., 2014; Reid et al., 2019). However, advances in our understanding of connections between geomorphological, hydrological and ecological processes across the riverscape require a new approach for fluvial characterization that can capture many variables concurrently and be conducted at scales relevant to key processes and their interactions (Beechie et al., 2010). These spatial scales are often intermediate in length (in the order of kilometers), domains over which continuous, high-resolution characterisation of channel conditions has traditionally been a challenge due to the time and cost constraints of ground-based survey methods (Fausch et al., 2002). Over the past decade, the use of remotely piloted aircraft (RPAs) has helped overcome this challenge through the collection of high-resolution imagery over a range of scales for evaluation of stream bed topography (e.g. Tamminga et al., 2015; Woodget and Austrums, 2017), bathymetry (e.g. Kasvi et al., 2019), and ecological parameters (e.g. Roncoroni and Lane, 2019). However, much of this work has been limited to larger systems, where the forest canopy has a limited impact on obstructing view of the channel. By contrast, smaller streams can be more prone to an obstructed view of the channel from dense forest canopies. Given the importance of in-stream wood for channel structure and function, particularly in smaller systems (Hassan et al., 2019), we consider the classification by Hassan et al. (2005) for small to intermediate streams in the Pacific Northwest as those where the ratio between wood length to bankfull channel width is close to or greater than one and the ratio between wood piece diameter and bankfull depth is close to or greater than one (see Table 2 of the paper for more

details). Streams on the intermediate side of this spectrum, where the ratio between bankfull channel width and wood length is close to one, differ from larger systems as they can be greatly influenced by wood delivered to the channel (Wohl and Scott, 2017). These channels are often overlain by dense forest canopies and are poorly suited to observation from above the forest canopy. This limitation has historically excluded a large fraction of river network length from RPA-based surveys.

The primary objective of this paper is to develop and test a methodology based upon spatially continuous RPA-derived data in order to objectively classify morphological units and characterise scales of variability in small, forested rivers under dense forest canopies. The variables considered for the classification include channel slope, water depth, and grain size characteristics, all of which reflect larger basin-scale controls on channel morphology (Buffington and Woodsmith, 2003), and are easily extractable from RPA imagery. Channel slope is a key variable to consider, as it has been shown that there is a general

progression of channel morphologies from pool-riffle, plane-bed, and step-pool to cascade morphologies with increasing slope (Montgomery and Buffington, 1997). Water depth metrics are important for discriminating between pool areas and other shallow water environments. Finally, grain size is a key variable as there tends to be a coarsening in bed material from glides and pools to riffles and runs (Garcia et al., 2012). In an effort to improve the characterization of these channels, we developed a new framework to map and classify channel attributes through the use of RPA-based data collection under forest canopies. To

build this framework, this paper aims to address the following research questions:

1. What are the capabilities and limitations of a survey approach using sub-canopy RPA flights to characterise channel attributes in small, forested streams?

2. Can spatially-continuous RPA-derived measurements be used to objectively characterise patterns in channel morphology?

3. What is the spatial extent of data collection necessary to capture the primary variability in geomorphic channel attributes?

To address these questions, a sub-canopy RPA survey was conducted along approximately 3.0 km of channel in Carnation Creek, B.C., a small coastal stream located on western Vancouver Island. This site serves as a valuable testing area due to the abundance of complementary data available through annual total station surveys of the channel's study sections and longitudinal profile survey data across the channel's lower 3.0 km (Tschaplinski and Pike, 2017; Reid et al., 2019).

## 2    Study area

This research was conducted along Carnation Creek, a small gravel bed river located on the southwest coast of Vancouver Island, B.C. (Fig. 1). The watershed has been the site of a long-running fish-forestry interactions study focusing on the effect of different logging treatments on watershed response (Tschaplinski and Pike, 2017). The channel mainstem is approximately 8 km long and has a drainage area of 11.2 km$^2$ (Tschaplinski and Pike, 2017). The focus of research is along the lowermost 3.0

90    km of the channel, which possesses a low gradient (0.5–1%) and is dominated by a pool-riffle channel morphology. Upstream, the channel narrows into a canyon (Fig. 1) which contains a predominantly step-pool morphology and gradient above 5% (Reid

et al., 2019). The average bankfull width ($w_b$) of the lower channel is close to 15 m. The channel is located within the Coastal Western Biogeoclimatic Zone, common along coastal regions of the Pacific Northwest (Hartman et al., 1982). Visual estimates suggest that over 50% of the channel is hidden below a dense forest canopy composed of both coniferous and deciduous tree species. The riparian vegetation consists of a variety of tree species including western hemlock (*Tsuga heterophylla*), Amabilis fir (*Abies amabilis*), western redcedar (*Tsuga plicata*), Sitka spruce (*Picea sitchensis*) and red alder (*Alnus rubra*). The height of the riparian canopy is variable, between approximately 15 and 40 m. The riparian forest floor is composed of a variety of ferns and shrubs, such as salmonberry (*Rubus spectabilis*), sword fern (*Polystichum munitum*), trailing blackberry (*Rubus ursinus*) and thimbleberry (*Rubus parviflorus*) that may provide some cover to the channel. The environment is typical of the Pacific Northwest: precipitation rates are high and dominated by rain (between 2,900 – 5,000 mm/year), the majority of which falls during the autumn and winter months (Tschaplinski and Pike, 2017). Streamflow ranges from 0.1 m$^3$/s to 64 m$^3$/s in fall and winter months, (Tschaplinski and Pike, 2017), and is often very low ($< 0.01$ m$^3$/s) for extended periods in the summer (Reid et al., 2020). Frequent storms in the winter months lead to multiple floods per year that are capable of mobilizing gravel in the system, with bankfull discharge between 20 and 30 m$^3$/s (Haschenburger, 2011).

The processes governing the morphological and hydraulic conditions in Carnation Creek are irregular in both time and space, creating a great deal of heterogeneity along the channel. Sediment is predominantly delivered from episodic landslides and debris flows located in the upstream half of the watershed, while large logjams intercept delivered material and lead to spatially variable sediment textures and morphological features (Reid et al., 2019). The sediment texture of the bed varies from small gravels near the stream outlet to coarser cobbles and boulders in the steeper canyon reach, but varies substantially over short distances (Reid et al., 2019). The bed surface and subsurface sediment textures are similar, representative of systems that experience comparatively high sediment supply conditions (Hassan et al., 2006).

Detailed morphological data have been collected through annual topographic surveys in eight study sections (SAs 2–9), seven of which (SAs 2–8) are located downstream of a canyon (termed the 'canyon reach', see Fig. 1). The eighth study section (SA9) is located away from the others, upstream of the canyon. The lower study sections are 300–500 m apart and 5–10 $w_b$ (50–150 m) in length (Reid et al., 2019).

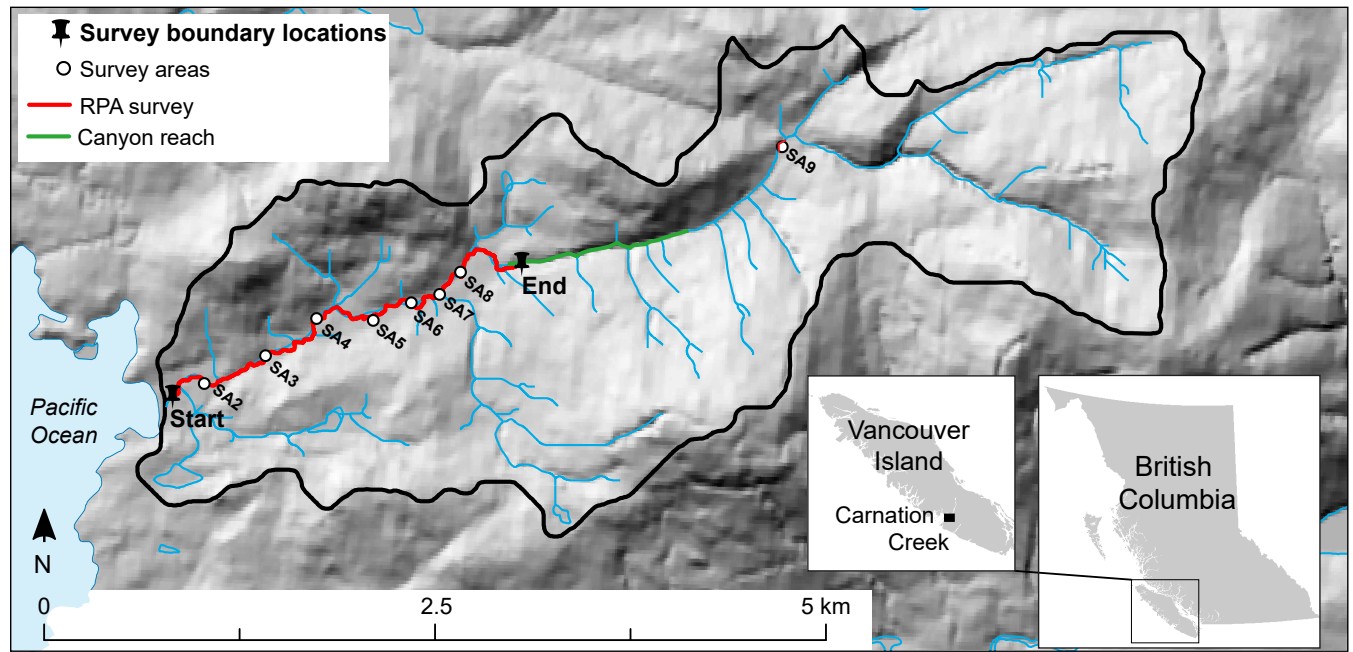

**Figure 1.** The Carnation Creek watershed, located on the south-west coast of Vancouver Island. The RPA survey extent is shown as a red line. An additional site (SA1) located in the channel estuary was active until the late 1980s, but has since been abandoned and was not included in this survey. Note that the RPA survey also included coverage of SA9, upstream of the other sites.

## 3 Methods

### 3.1 Remotely piloted aircraft survey

In July of 2018, approximately 3.0 km of channel was surveyed, with coverage extending from just upstream of the river mouth to the downstream limit of the canyon reach (Fig. 1), as well as over most of the SA9 study section. SA9 is farther upstream and possesses smaller channel dimensions with a closed canopy that provides cover to the channel, and therefore serves as a challenging test site to navigate and evaluate the coverage attainable with the RPA. Total survey time was approximately 12 full days, including flights over SA9. The RPA survey involved low-level flights (5–15 m above ground level) conducted in tandem with placement of ground control points (GCPs) on the dry exposed bars and checkpoints on both the exposed and submerged bed. Flights were operated manually below the canopy to have an unobstructed view of the channel bed, and because low-hanging vegetation and the forest canopy made pre-planned flights impractical. The flights were undertaken with a DJI Phantom 4 Advanced RPA, a consumer-grade RPA which contains a camera with a focal length of 8.8 mm (24 mm in 35 mm format equivalent) and a field of view of $84°$. To obtain sufficient overlap between images, frames were acquired at two second intervals while moving at approximately 1 m/s horizontal velocity.

Due to flight obstacles (low-hanging branches, fallen trees, etc.), sightline obstructions, RPA battery life, and other practical
survey challenges, the 3.0 km of channel was divided into roughly 80 segments, covered by 300–1,000 photos each. Each
segment was initially flown following flight lines parallel to the channel direction, with imagery collected at 90° relative to
the bed plane. While this in-flight photography strategy captured much of the channel, bank areas were often obstructed from
overhead view by low-elevation shrubs, ferns, and brambles. To capture these obscured channel areas, each segment was
flown with oblique and convergent imagery. 'Oblique imagery' refers to frames captured with a camera angle differing from
135 bed-perpendicular, while 'convergent' refers to images capturing the same bed area but from different approach directions.
This approach to image collection is likely advantageous in streams where riparian vegetation may prevent the RPA from
flying directly over the bank, and has led to improvements in the quality of survey outcome in several studies (Wackrow and
Chandler, 2011; James and Robson, 2014; Harwin et al., 2015). To collect this type of imagery, the RPA camera was tilted at a
low angle (20–30° from vertical, see Figure 2 a) and a flight path parallel to the banks was taken (see Figure 2 b).

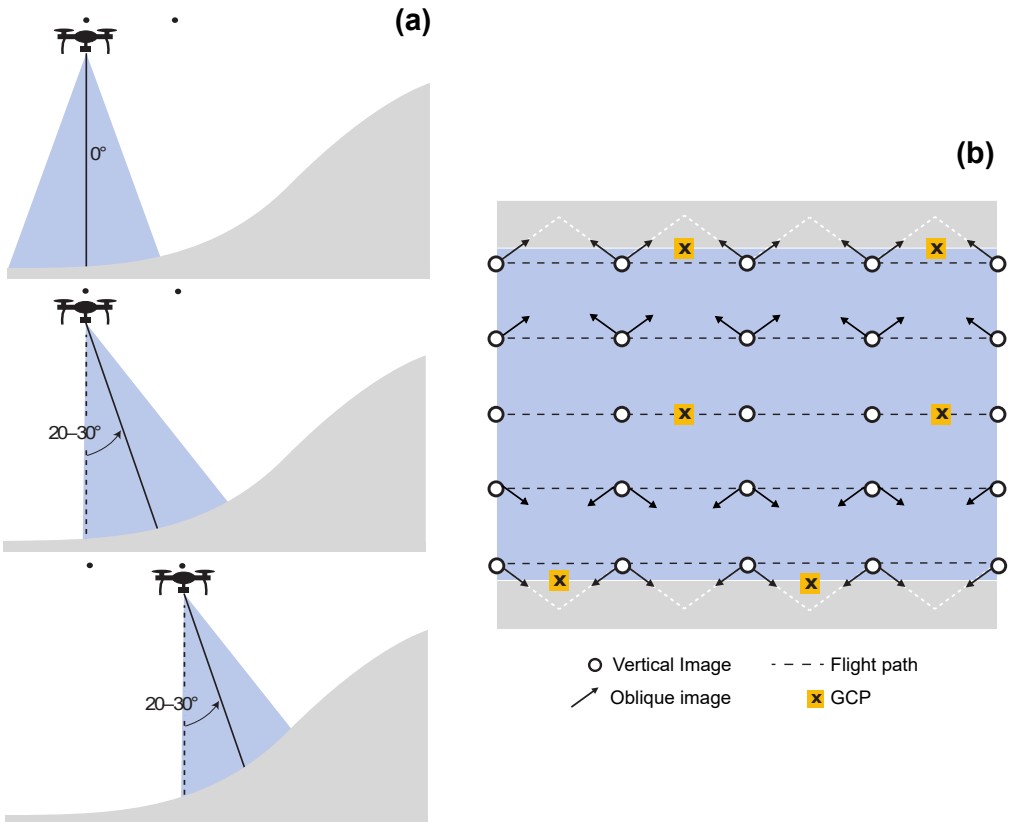

**Figure 2.** (a) Example partial channel cross-section showing the oblique angles of the RPA's camera (solid black line) for image acquisition
of the banks. To characterise the channel banks, the camera was tilted 20–30° from vertical. (b) Plan view of the flight path of the RPA with
the parallel flight lines shown as dashed lines. The outlined circles show the locations of a vertical image, and the arrows show the horizontal
orientation of the camera towards the channel banks for the oblique images described in (a).

A minimum of ten GCPs (composed of approximately 0.1 m x 0.1 m ceramic tiles with a central X marking the surveyed location) were placed along dry exposed bars in each of the 80 channel segments to provide precise image georeferencing, with additional tiles positioned on the dry exposed bars and below the water surface in order to serve as independent checkpoints, to assess the accuracy of the model outputs. The majority of the GCPs were distributed in a zig-zag fashion along dry exposed bars in the periphery of the channel segments, with a smaller number situated towards the centre. This configuration provided a

balance between the suggested distributions of GCPs found in previously published studies (Harwin et al., 2015; Agüera-Vega et al., 2016; Tonkin and Midgley, 2016; Sanz-Ablanedo et al., 2018). All GCPs and checkpoints were surveyed with a Leica TPS 1100 total station. Open survey traverses were tied into benchmarks previously established in the study sections, and then an affine transformation applied to georeference the points in the XY-plane. The average offset between the benchmark elevations of the local open traverse and their known reference elevations were then used to georeference the points in the

Z-plane. Errors were typically 0.02 m in the XY-plane, and 0.01 m in the Z-plane.

### 3.2   Basedata extraction

Channel elevation, bathymetry and grain size were extracted from the RPA imagery to aid in the classification of channel unit morphology. A digital elevation model was generated of the site using the software Agisoft PhotoScan Professional (AgiSoft, 2017) to generate georeferenced dense point clouds of each surveyed channel segment. As riparian vegetation often obstructed parts of the channel bed and introduced errors when digital elevation models are generated from point clouds (Tamminga et al.,

2015), the Cloth Simulation Filter (Zhang et al., 2016) from the open source software Cloud Compare (Cloud Compare, 2017) was employed. This tool inverts the point cloud and generates an interpolated surface analagous to 'draping' a simulated cloth over the ground surface to approximate the terrain of an obscured area (Zhang et al., 2016). Following visual inspection of the filtered result, a cloth resolution of 0.1 m and maximum distance between 0.5 to 1.0 m was found to adequately filter the

bed points. The cloth resolution represents the horizontal spacing between points in the cloth, whereas the maximum distance represents the threshold used to classify ground and non-ground points based on the distance between the original cloud and cloth.

    The elevations of submerged channel bed areas are often overestimated due to the refractive effect of overlying water (Dietrich, 2017). To correct for this effect and to develop accurate bathymetry, a corrective Python script developed by Dietrich

(2017) was employed. By determining the distance from a generated water surface mesh to the estimated bed elevations in the point cloud below, the corrected water depth for a location could be calculated as a function of the multiple viewing angles used to observe each point. The method requires that the water be clear such that the channel bed can be captured. The low flow conditions present at the time of the survey resulted in clear water that permitted viewing of the channel bed. Removal of overhanging vegetation using the Cloth Simulation Filter in Cloud Compare, and subsampling the point clouds and resulting

DEMs to a spacing of 0.02 m using the minimum elevations in the point cloud, helped to ensure that the refraction correction was based on channel bed points and not on overhanging vegetation points that may have been incorporated in the point cloud.

    Grain size estimates of the exposed bed were important to extract from the bed imagery, as patterns in sediment texture often follow patterns in channel morphology and are frequently discussed in classification schemes (e.g. Montgomery and

Buffington, 1997). Grain size estimates were acquired by establishing a relationship between the roughness of the point cloud

for 22 training sites and their median grain size ($D_{50}$) (see method described by Woodget and Austrums, 2017), a metric often of interest to river managers. Each roughness sampling site was approximately 1 m$^2$ and imagery was captured for photo-sieving by hovering the RPA approximately 2 m above ground level. Using an in-house photo-sieving program based in Matlab (Matlab, 2017), the grain size distribution of each training site was determined. The program loads the image, prompts the user to scale the image, and then overlays a grid with 50 nodes prompting the user to measure the B-axis of grains falling below

a grid node. Point clouds for each sample site were then extracted from the georeferenced point cloud that was developed for the study section that they fell within, and a roughness value for each point estimated using the roughness tool in Cloud Compare. DEMs were then developed for each roughness site at a 0.02 m resolution and a mean roughness value for each DEM determined using R. A linear model then was then fit between each sample's $D_{50}$ (Fig. 3) and its mean roughness value. Using a 1 m$^2$ moving window (which approximates the size of the training sites), grain size was then estimated across the exposed

bed as described by Woodget and Austrums (2017).

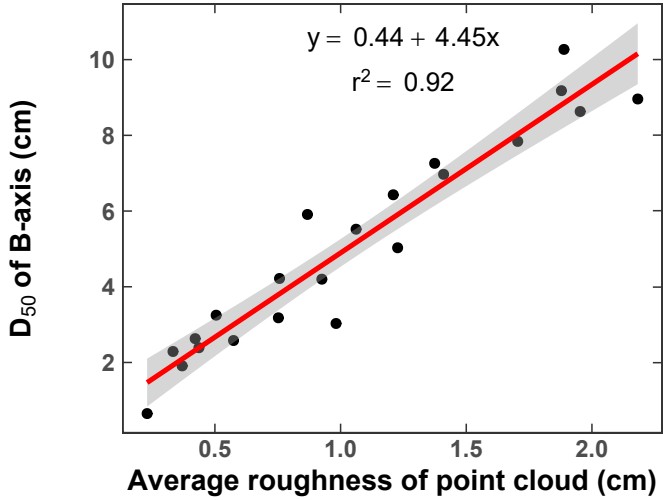

**Figure 3.** Predictive grain size relationships between the median surface sediment calibre ($D_{50}$) and the average roughness value of the training sites as determined from RPA-derived bed surfaces.

### 3.3 Selection of channel variables

To classify the channel along the longitudinal profile, the thalweg was first identified using the River Bathymetry Toolkit (RBT), an ArcMap add-in (McKean et al., 2009). The thalweg was used as a standardized location along which observations would be extracted at fixed 1 m intervals to provide a smooth transition in channel unit morphologies. To characterise patterns in channel

unit morphology, five variables were extracted: the hydraulic radius ($R_h$), median grain size ($D_{50}$), local bed ($S_l$) and water surface slope ($S_{ws}$), and the reach bed slope ($S_r$). These variables were chosen as they are straightforward to extract from

the data (a key requirement for a rapid classification scheme), and because they reflect larger basin-scale variables relevant to channel form, such as geology, climate and land-use. To provide a measure of grain roughness across the channel, the average $D_{50}$ of the dry exposed bars in a 0.5 m buffer around each sampling location's cross-section was extracted. The local slopes of the bed and water surface (extracted from point clouds of the water surface mesh that were generated from the Dietrich (2017) routine) were calculated for each sampling location by fitting a linear model through observations in a 15 m window around each sample site. This was repeated for the reach-scale bed slope using a 45 m window. Together these variables summarize the channel form ($R_h$ and $S$) and roughness of each cross-section. Cross-sections where the channel banks were not discernible (due to channel obstructions or dense low-lying vegetation) were excluded from the analysis. Exclusion of these cross-sections, along with segments of the channel the RPA could not access, comprised approximatley 25% of the channel's thalweg.

## 3.4 Analysis

Following the extraction of the five channel variables, a principal component analysis (PCA) was applied to determine which variables were important for characterizing channel unit morphology, and a k-means clustering approach was then used to classify the PCA results into morphological units. To implement the PCA and k-means clustering, the package 'stats' in R was employed (R Core Team, 2018). The general objective of a PCA is to reduce the number of dimensions in a dataset that contains interrelated variables while describing the maximum amount of variation present (Jolliffe, 2002). Because the dataset was multi-dimensional with five variables over 2,362 sampling sites, a PCA was an appropriate tool to help simplify and extract patterns in the data, a prerequisite for k-means clustering. The PCA was run and then three of the five components were retained for further analysis, which together explained approximately 79.0% of the variation in the dataset, an appropriate cut-off according to Jolliffe (2002).

Following the PCA, the k-means clustering algorithm was run to identify groupings that may have been present in the data along its first three components. A k-means clustering algorithm is an unsupervised classification that assigns observations from $n$ dimensions to clusters that allow the within-cluster sum of squares to be minimized (Hartigan and Wong, 1979). Following guidelines for the method described by Flynt and Dean (2016), six clusters were chosen to group the dataset, a value which is in reasonable agreement with the number of channel unit morphologies one may expect at Carnation Creek.

Following clustering of the cross-sectional variables, the mean values of each channel variable for each cluster were examined and one of the following morphological units attributed to each cluster: pool, riffle, coarse riffle (riffle$_C$), glide, run or plane-bed. The units were assigned to clusters based on obvious features (e.g. shallow water slopes and greater depth for pools, negative pool exit slopes for glides, and steeper pool entry slopes for runs) and criteria presented in Church (1992), Anonymous (1996), and Buffington and Woodsmith (2003). These criteria are described in Table 1. The resulting assignment of morphologies to clusters leads to a continuous classification of morphological units found along the study reach at 1 m intervals, and provides insight into the survey extents necessary to adequately capture the heterogeneity of the system.

To characterise the diversity of morphological units across the stream, a moving analysis using the Shannon diversity index (Shannon and Weaver, 1964) was conducted. This index provides a measure of the abundance and evenness of a property in an area (Lloyd and Ghelardi, 1964). While this index is often calculated with regard to species types in ecology, the approach can

**Table 1.** Average values for variables from morphological units found in previously published studies.

| Morphology | $S_{Church}$ (m/m)[a] | $S_{Hogan}$ (m/m)[b] | $S_{Buff.}$ (m/m)[c] | $D/d_{Church}$ (m)[d] | $D/d_{Hogan}$ (m)[e] |
|---|---|---|---|---|---|
| Riffle | 0.02 | 0.005–0.015 | 0.001–0.02 | <1.0 | 0.1–0.3 |
| $Riffle_C$ | - | 0.015–0.03 | - | - | 0.3–0.6 |
| Plane-bed | 0.02-0.04 | 0.03–0.05 | 0.01–0.04 | ∼1 | 0.6–1.0 |

[a] Slope values published from Church (1992)

[b] Slope values published from Anonymous (1996)

[c] Slope values published from Buffington and Woodsmith (2003)

[d] Relative roughness values published from Church (1992)

[e] Relative roughness values published from Anonymous (1996)

also be applied to morphological units, similar to the work of Harris et al. (2009). To calculate index values, the proportion of each morphological unit in an area is multiplied by the natural logarithm of the proportion. These values are then summed for all the morphological units present in an area. In order to apply the method to the Carnation Creek data, the index values are first calculated by iteratively dividing the channel into segments based on window sizes ranging from 15–750 m in length (at 15 m intervals). For each iteration, the abundance of each morphological unit in each channel segment was determined. Using the 'vegan' package in R, the Shannon's diversity index of each channel segment was then calculated.

To determine the spatial scale required to capture the heterogeneity of the channel, diversity metrics were first calculated for each iteration (using an increasing window size ranging from 1–50 $w_b$ in length), and then the standard deviation of all the diversity values for each iteration was calculated. For example, for the first iteration, diversity metrics were calculated across the channel based on 15 m segments. A standard deviation value was then calculated from all the diversity metrics for the iteration. As sample size increases, the standard deviation of the diversity index from the channel segments would be expected to tend towards an asymptote. The length scale required to approach this asymptote can therefore be interpreted as the scale beyond which diminishing returns arise in variability captured.

## 4   Results

### 4.1   Accuracy of the RPA survey

The channel-averaged vertical survey error was estimated by calculating the root-mean-square-error (RMSE) and the mean error (ME) of differences between the elevations of check points collected with the total station survey and those estimated from the DEMs. The RMSE provides a measure of the spread of the squared residuals whereas the ME provides a measure of any potential positive or negative bias to the data, and are similar to other metrics used to evaluate RPA survey performance

(e.g. Tamminga, 2016). The overall spread of this error and summary statistics are illustrated in Fig. 4. Vertical errors of the exposed bed points were found to be 0.093 m and 0.025 m for the RMSE and ME, respectively (n = 1,203), and similar values were obtained for the submerged bed points (RMSE = 0.11 m, ME = 0.025 m, n = 521). As shown in Fig. 4, the majority of the errors for the submerged points were close to 0. However, factors such as shadows from the riparian vegetation and reflections from the canopy may have influenced the success of the refraction correction (Dietrich, 2017).

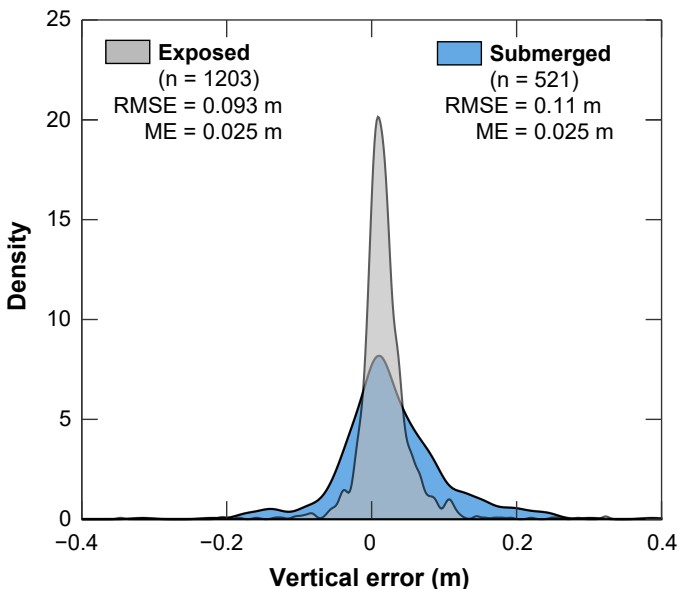

**Figure 4.** Density plot displaying the distribution of vertical errors between the modelled and field measured elevations. Summary statistics (root-mean-square-error (RMSE) and mean error (ME)) are provided for both the exposed and sumberged checkpoints.

## 4.2   Coverage with the RPA survery

In order to evaluate the coverage extent obtainable with the sub-canopy RPA survey, the RPA-based results were compared to channel boundaries delineated with a total station in the eight established study sections (see example in Fig. 5). When including side channels, which were generally difficult to access with the RPA due to dense sub-canopy vegetation, it was possible to capture approximately 80% of the delineated study sections, a value which increased to 87% when side channels are excluded. When examining individual study sections that contained side channels, coverage ranged from a low of 54% in SA4, to a high of 89% in SA9. Generally, narrow (width < 3 m) side channels could not be effectively surveyed, but oblique imagery was advantageous in situations where a clear flight path was present alongside an obscured channel area (Fig. 6). Similarly, bank top elevations were difficult to capture in most locations due to understory vegetation obscuring the ground surface. The inclusion of bathymetric calibration greatly increased the area over which bed topography could be estimated (Fig. 6).

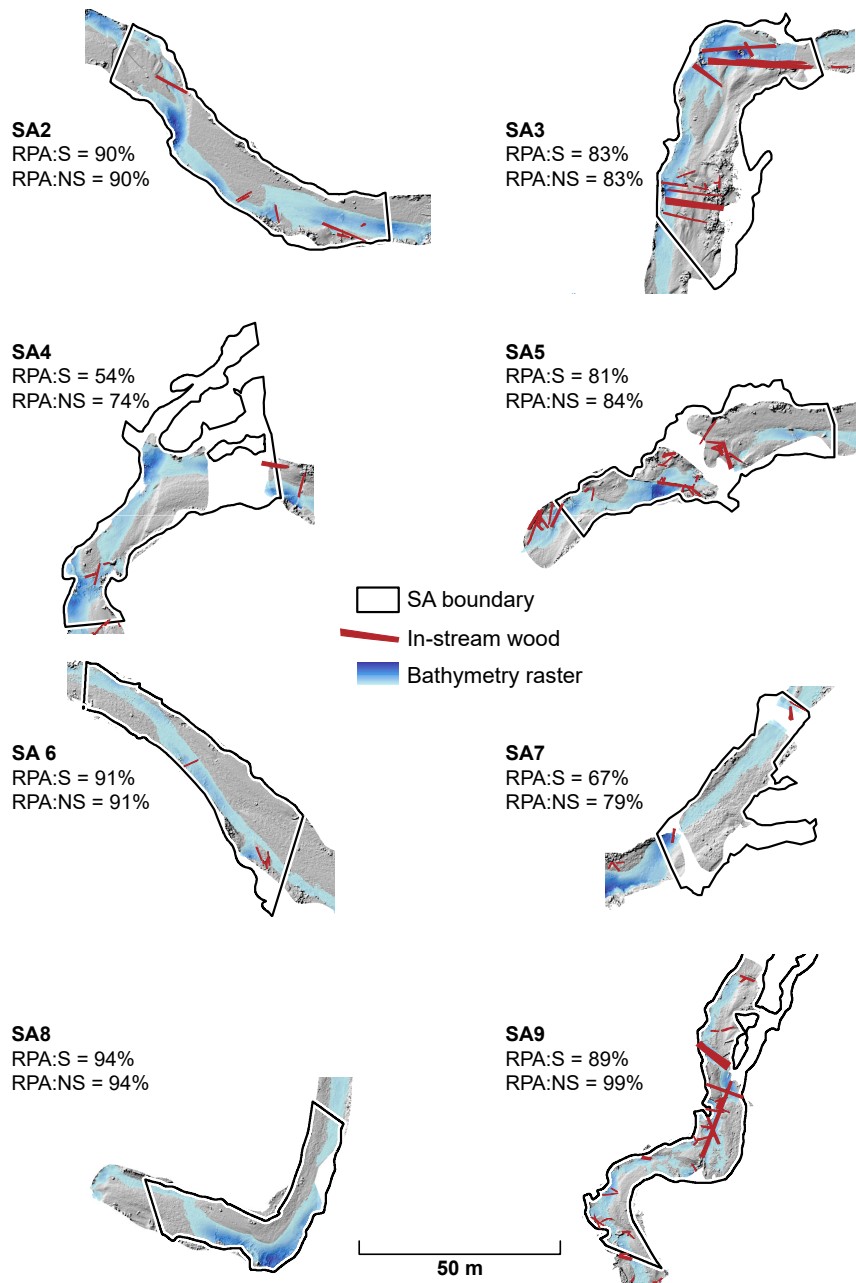

**Figure 5.** RPA coverage in comparison to the study section boundaries for SAs 2–9. Percentages of the study section covered with the RPA relative to the total station are based on whether the reference boundary included side channels (RPA:S) or just the main channel (RPA:NS). In-stream large wood (LW) was manually digitized using the DEMs and orthomosaics of the study sections. Pieces of wood (larger than approximately 0.1 m in diameter and 1 m in length) were digitized individually, whereas log jams were digitized as polygons as a result of difficulties in identifying individual pieces embedded within jams.

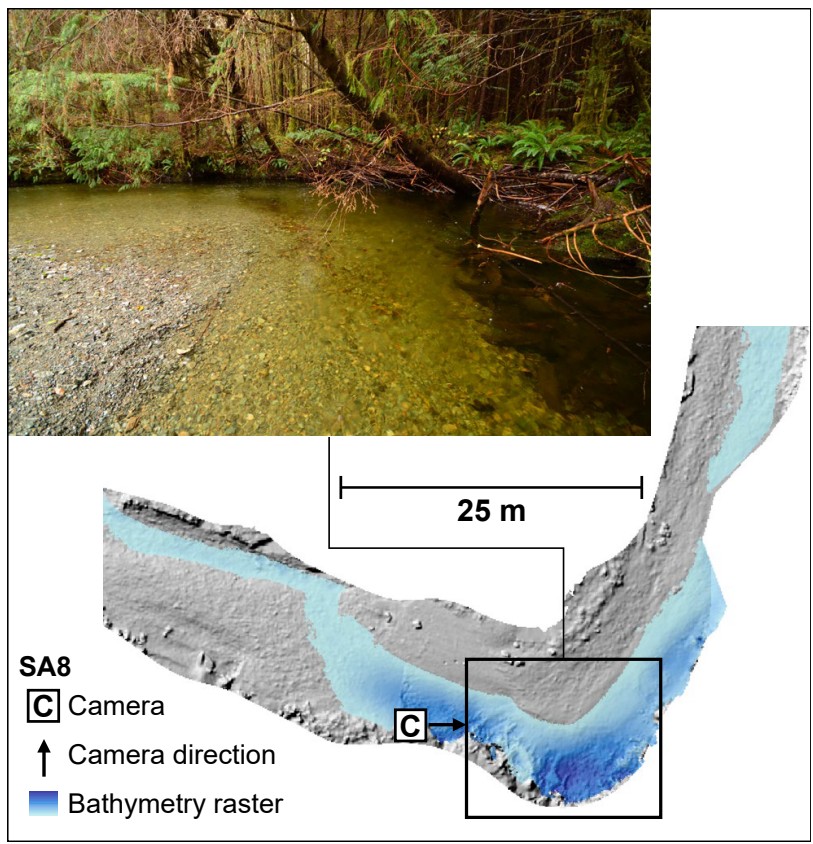

**Figure 6.** Coverage of a deep pool in SA8 under dense riparian vegetation. Note that the photo was taken in the autumn prior to the RPA survey, when the water level was higher than it was during the RPA survey. Photo courtesy of Iain Reid.

### 4.2.1 Principal component analysis, clustering analysis, and channel classification results

The first three components from the PCA explained approximately 80% of the variation in the data, with components one, two and three reflecting 45.11%, 19.3% and 14.6% of the variation, respectively. The first component is dominated by $S_r$, $D_{50}$ and $S_{ws}$, the second by $R_h$, and the third by $S_l$ and $D_{50}$. After running the k-means clustering algorithm using six groupings on the first three components, these patterns were evident along the axis of the biplot (Fig. 7). For each cluster, the mean of each variable was calculated and the likely morphological unit corresponding to the cluster estimated from these values (Table 2). Moving from left to right along the first dimension (Fig. 7) there is a shift from unit morphologies with lower bed and water surface slopes and finer bed sediment to those with steeper gradients and coarser material. This appears to represent a transition from pool to riffle unit morphologies along the first component. Overall, distinctions between most channel attributes arising from the clustering are clear and lead to relatively unambiguous classification of morphological units (Table 2). Within the riffle channel unit, the classification also captures a distinction between riffle unit morphologies with slightly coarser bed material, defined here as 'riffle-coarse' (riffle$_C$, see Anonymous, 1996). When examining the second component (y axis of Fig. 7),

hydraulic radius ($R_h$) decreases from top to bottom, as indicated by the transition from lower-velocity pool to higher-velocity glide unit morphologies, with remaining unit morphologies possessing intermediate $R_h$ (Fig. 7).

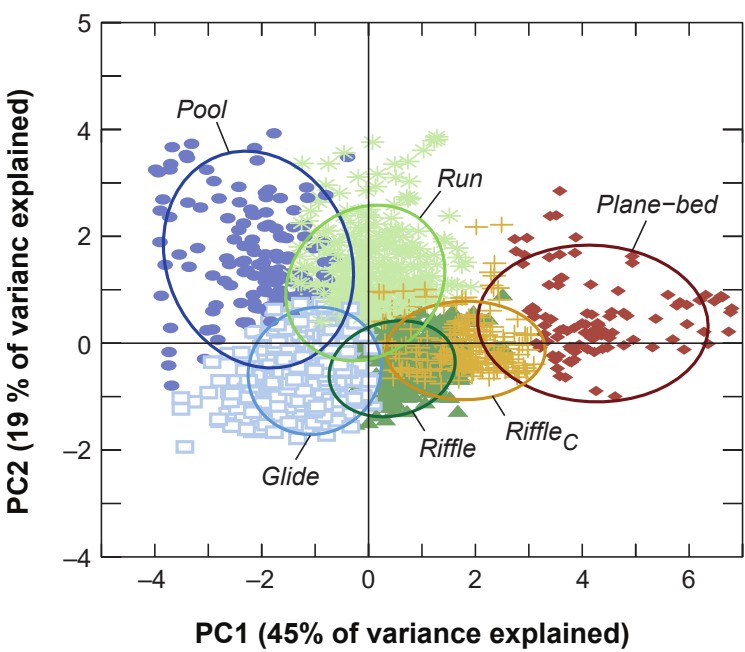

**Figure 7.** Biplot of each observation along the first two principal components PC1 and PC2. The groupings from the k-means clustering analysis are colour-coded and their centroid outlined.

Pools, riffles, glides and runs are relatively well distributed along the surveyed length of channel (Fig. 8). However, plane-bed and coarse riffle morphological units are mostly located near the upstream limit of the survey extent in this region. This area represents the outlet and downstream entrance of the canyon reach, where steeper gradients and coarser sediment are found. This is highlighted in Table 2, which shows that on average these morphological units are located 3160 m upstream, with steep reach scale gradients of 0.042 m/m and coarse material with an average $D_{50}$ of 0.082 m. Similarly, the coarse riffle morphologies were located approximately 2980 m upstream on average, with relatively steep gradients and coarse material (reach scale slope = 0.024 m/m and $D_{50}$ = 0.067 m). By contrast, the average positions of the riffle, glide, run and pool morphologies were approximately 1500 m, midway along the channel's profile, indicating that these morphological units are distributed over a greater length of channel. Grain size was generally similar between these morphologies, except for the riffle unit, which was slightly coarser with a $D_{50}$ of 0.041 m. Pools were the deepest, with average water depths of 1.04 m and near zero water surface slopes, whereas riffles were the shallowest with average water depths of 0.13 m and relatively steep water surface and reach scale bed slopes. Glides and runs were intermediate between these morphologies, with glides often retaining negative local bed slopes, corresponding with the exit of pools, and runs with large positive local bed slopes, corresponding with the entry of pools.

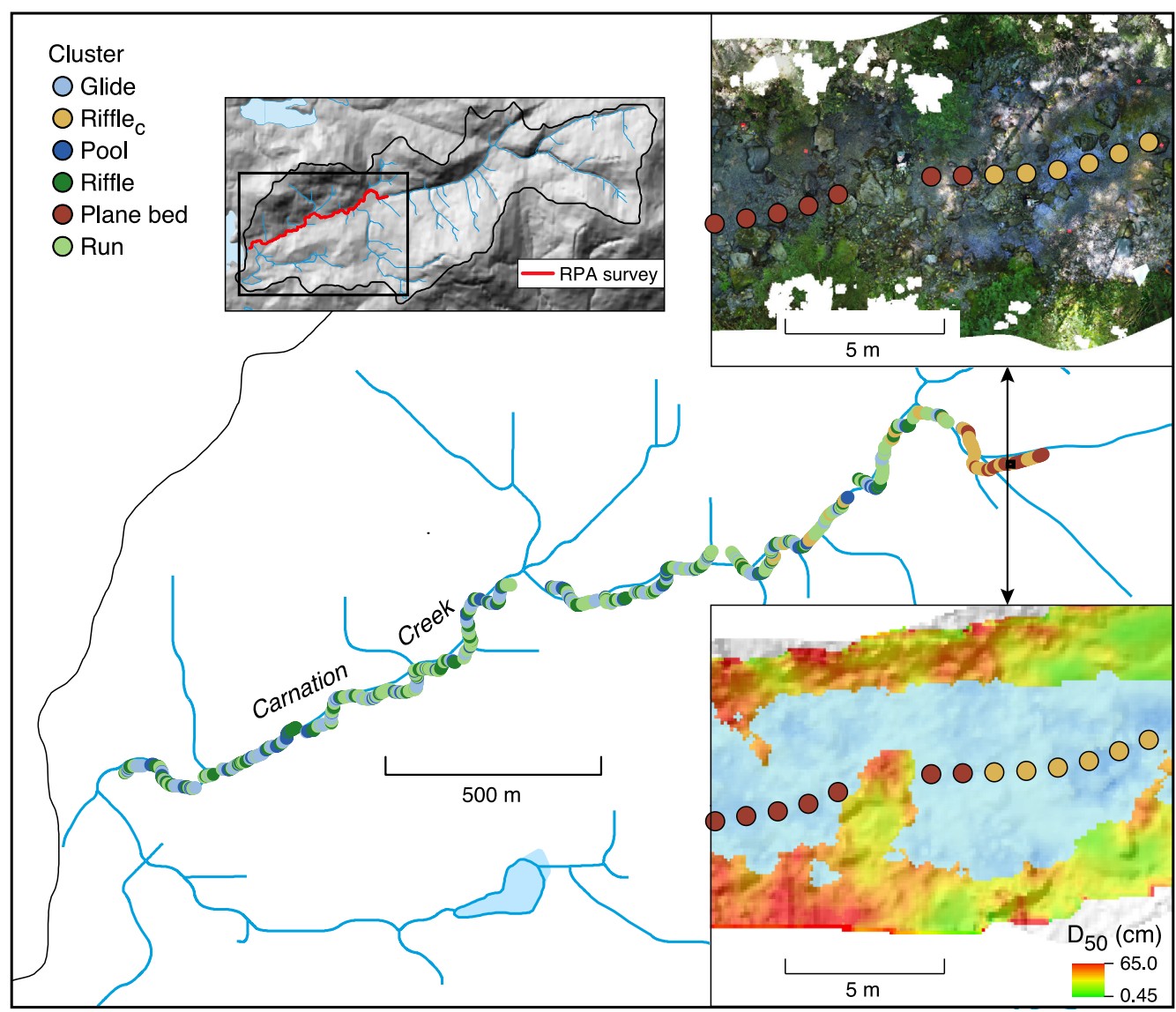

**Figure 8.** Distribution of morphological units along the surveyed reach of Carnation Creek. At approximately 2,500 m upstream, there is a marked change in channel unit morphologies from pool, riffle, run and glides to much steeper and shallower channel morphological units.

**Table 2.** Means of channel variables for each cluster.

| Cluster | $l$ (m)[a] | $d$ (m)[b] | $R_h$ (m/m)[c] | $S_l$ (m/m)[d] | $S_{ws}$ (m/m)[e] | $S_r$ (m/m)[f] | $D_{50}$ (m)[g] | $W$ (m)[h] |
|---|---|---|---|---|---|---|---|---|
| **Riffle$_C$** | 2980 | 0.16 | 0.12 | 0.018 | 0.018 | 0.024 | 0.067 | 4.13 |
| **Plane-bed** | 3160 | 0.20 | 0.14 | 0.054 | 0.047 | 0.042 | 0.082 | 3.47 |
| **Riffle** | 1650 | 0.13 | 0.090 | 0.027 | 0.016 | 0.012 | 0.041 | 3.65 |
| **Glide** | 1470 | 0.28 | 0.16 | -0.020 | 0.003 | 0.003 | 0.037 | 4.99 |
| **Run** | 1435 | 0.61 | 0.35 | 0.044 | 0.005 | 0.016 | 0.039 | 4.94 |
| **Pool** | 1420 | 1.04 | 0.60 | -0.031 | -0.004 | 0.000 | 0.037 | 5.99 |

[a] The midpoint of the longitudinal span where the morphological unit occurs

[b] Thalweg depth

[c] Hydraulic radius

[d] Local slope

[e] Water surface slope

[f] Reach-average slope

[g] Median grain size

[h] Wetted channel width

## 4.3 Assessment of the channel classification

To assess the accuracy of the clustering algorithm, 100 locations along the surveyed length of channel were randomly selected and visually assigned to either glide, pool, run, riffle, riffle$_C$, or plane-bed morphological units. These values were then compared to the morphological units predicted by the PCA. A summary of agreement between the PCA and visual classification approach is shown in Table 3. On average, 85% of sampled locations received the same morphological unit assignment between the two approaches, with riffle areas showing the lowest agreement (72%) and plane-bed areas the highest (100%). Overall, the classification matches the typical expected progression of channel unit morphologies in a pool-riffle system, as is shown in Fig. 9 . The exit of the pool is classified as a glide, with negative bed surface gradients. As gradient increases we see shallow riffle unit morphologies that meld into a deeper run at the entry of the pool (Fig. 9). It is likely that much of the disagreement can be attributed to 'transition' morphologies, which most classification schemes are unable to capture or define.

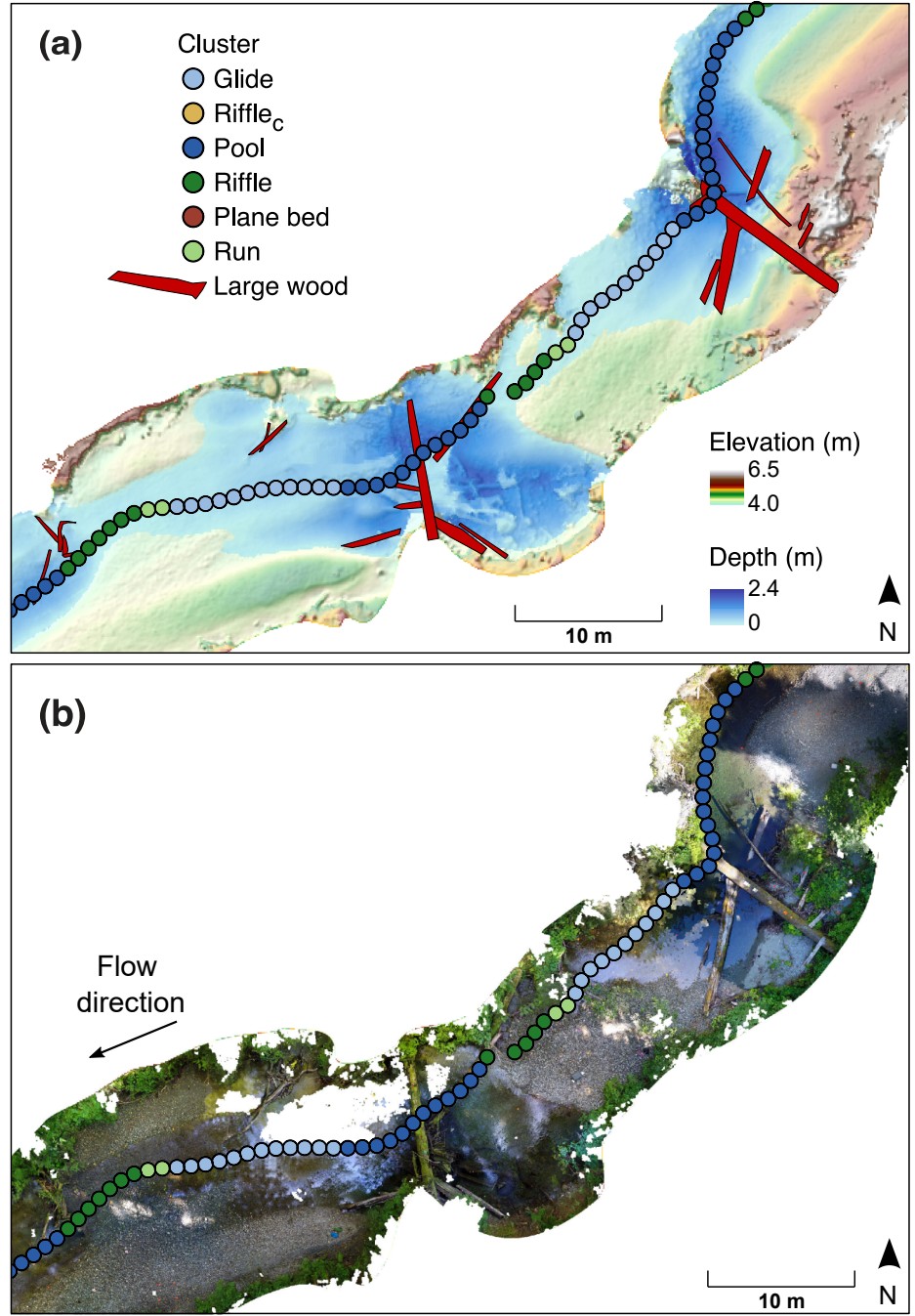

**Figure 9.** Example sequence of morphological units predicted from the k-means clustering algorithim. The figures show the transition from riffles to pools in a heterogeneous section of channel overlaid on (a) a DEM and (b) an orthomosaic. Note the hole at the downstream pool (b) which is due to overhanging vegetation that prevented stitching of the orthomosaic for this area. By contrast, in (a) this vegetation was removed using the Cloth Simulation Filter in Cloud Compare (Zhang et al., 2016), resulting in a clear DEM of the bed.

**Table 3.** Accuracy assessment of morphological unit classification using k-means clustering.

| Morphological unit | % Correctly classified |
|---|---|
| *Riffle$_C$* | 78 |
| *Riffle* | 72 |
| *Plane-bed* | 100 |
| *Glide* | 97 |
| *Run* | 85 |
| *Pool* | 80 |
| *All* | 85 |

## 5  Discussion

### 5.1  Utility of sub-canopy RPA surveys for small, forested streams

The results of this study provide a precedent for using RPAs to characterise morphological units in small, forested streams below the forest canopy. This approach provides several advantages over traditional ground-based surveys. We have demonstrated that over 12 field days, nearly three kilometers of a small forested channel could be surveyed with an estimated coverage rate of 80% (including side channels) at a greater spatial resolution and extent than most traditional ground-based methods allow. For example, the traditional total station-based surveys conducted in Carnation Creek typically result in point densities of 0.5–1.5 points/m$^2$, with 500–1000 points captured in a normal field day over a 70 m length of channel. In contrast, the average data acquisition rate with the RPA was 225 m/day, more than three times the length coverage from the total station approach, and at a much higher resolution. The DEMs and orthomosaics created from these images were of a very high resolution (0.02 m/ pixel) with survey uncertainty between 0.01 m (for dry areas) and 0.1 m (for submerged bed areas). This magnitude of error is comparable to values observed in other studies (e.g. Flener et al., 2013; Tamminga et al., 2015), and is similar to error achieved using traditional ground or GPS-based point surveys in the same channel (Reid et al., 2019).

Oblique imagery appears to provide good coverage of near-bank areas that are traditionally difficult to capture with vertical imagery, enabling the characterisation of low-velocity, near-bank channel areas which serve as critical fish habitat (Bjornn and Reiser, 1991). This additional imagery is generally straightforward to collect, but adds to the RPA power requirements and also increases survey time as a result of the need for additional flight passes. However, should repeat surveys be undertaken, a major reduction in survey time would be achieved through the installation of permanent ground control points. New RTK-GPS systems providing centimeter-level accuracy are also becoming available for consumer-grade RPAs, though signal attenuation through dense trees may reduce survey accuracy and limit their applicability for sub-canopy surveys.

While sub-canopy RPA surveys appear promising, certain environmental conditions and aspects of the survey approach continue to present limitations. First, the techniques for extracting the bathymetry may not be suitable for streams with turbid water that prevent observation of the submerged bed. While oblique imagery aided in characterisation of some bank areas, low elevation and dense riparian vegetation still pose a challenge for capturing bank topography in some locations, information which is necessary should the resulting survey be used for hydrodynamic modeling (Cienciala and Hassan, 2013) or to quantify bank erosion (Reid et al., 2019). In addition to bank vegetation causing obstructions, submerged areas with little texture and low-hanging branches (predominantly from riparian deciduous species) occasionally led to flight difficulties that prevented sufficient collection of imagery for photo-stitching. Therefore, these techniques may be most suited to small channels in relatively mature forests that have an open understory, and flights in winter months when foliage is absent may prove beneficial. In certain circumstances, a hybrid survey with both RPA and total station data could provide complete coverage, even in locations highly obscured by dense understory foliage. In spite of these limitations, however, the sub-canopy RPA survey approach appears to offer substantial improvements over traditional survey methods.

## 5.2 Assessment of the classification approach

The PCA-clustering classification approach appears to present a viable and less-subjective method for evaluating morphology at the channel unit scale, and incorporates a larger number of key variables than traditional methods. While some subjectivity remains in the interpretation of the k-means-derived clusters, examination of the classification from the PCA-clustering analysis revealed that there was good agreement between the characteristics of the morphological units derived from the clustering approach and morphological units identified visually (Table 3), with at least some remaining disagreement attributable to transition areas between morphological units. As shown in Table 4, the mean values of the variables for each assigned morphological unit are similar to reference values found for the slope, depth and grain size characteristics of similar channels classed in a number of other studies. Another advantage of the PCA is that it highlights the trends present in a dataset, rather than focussing on specific features. For example, anomalous areas where imagery may have had stitching issues due to poor coverage (e.g. SA5 in Fig. 5) would likely appear as noise, thereby having a minimal influence on the PCA.

Including frequently measured channel metrics in a PCA-clustering analysis, as was conducted in this study, provides a sophisticated means not only for relating physical conditions to channel form (as descriptive schemes tend to do), but for identifying which key variables impact the relationship. Such an analysis may provide a precursory understanding of key variables worthy of investigation in the development of process-based classification schemes. A challenge encountered by many classification schemes is that they often lack the generality to be applied in environments outside of those for which they were developed. For example, although Whiting and Bradley (1993) provided a strong process-based classification of channel form, it was intended for headwater channels, limiting its wider applicability (Buffington and Montgomery, 2013). Similarly, the approach to classifying channels proposed by Montgomery and Buffington (1997) has a clear process basis where the channel is partitioned into source, transport and deposition zones, but was developed for mountain drainage basins. While the classification approach proposed here is also based in a mountainous environment, the PCA-clustering technique allows for the identification of morphological units in any fluvial environment where sufficient variation in bed topography is

present. Unlike most classification schemes, identified clusters must be interpreted after the analysis to situate them within our conceptual understanding of river systems. While this consists of an additional step, it can provide opportunities to confirm our

understanding of field observations in river systems or to guide further investigation when unexpected patterns appear.

Finally, it should be noted that in order to characterise the geometry of the channel, the PCA approach relies on wetted variables, in contrast to flow-independent features like bankfull width or depth. When considering factors such as the needs of salmonids, the low flow conditions observed in late-summer may be of concern and will determine the connectivity and distribution of certain morphological units across the riverscape. Depending on the application, however, consideration of

flow-independent variables may be required, like the bankfull width or depth, which are less dependent on the particular wetted conditions observed at the time of the survey.

**Table 4.** Comparison of average values for variables of each morphological unit to those found in previously published studies. Values from this study are indicated in bold.

| Morphology | $S_{Church}$ (m/m)[a] | $S_{Hogan}$ (m/m)[b] | $S_{Buff.}$ (m/m)[c] | $S$ (m/m) | $D/d_{Church}$ (m)[d] | $D/d_{Hogan}$ (m)[e] | $D/d$ (m) |
|---|---|---|---|---|---|---|---|
| *Riffle* | 0.02 | 0.005–0.015 | 0.001–0.02 | **0.012** | <1.0 | 0.1–0.3 | **0.33** |
| *Riffle$_C$* | - | 0.015–0.03 | - | **0.024** | - | 0.3–0.6 | **0.41** |
| *Plane-bed* | 0.02-0.04 | 0.03–0.05 | 0.01–0.04 | **0.042** | ~1 | 0.6–1.0 | **0.42** |
| *Glide* | - | - | - | **0.003** | - | - | **0.13** |
| *Run* | - | - | - | **0.016** | - | - | **0.06** |

[a] Slope values published from Church (1992)

[b] Slope values published from Anonymous (1996)

[c] Slope values published from Buffington and Woodsmith (2003)

[d] Relative roughness values published from Church (1992)

[e] Relative roughness values published from Anonymous (1996)

## 5.3   Insight into scales of spatial variability

The results of calculating the standard deviation of the diversity metric for morphological units (Fig. 10 a) suggest that a window size of approximately 13–15 $w_b$ (175–200 m in length) is necessary to capture the dominant variability along the channel. The

inflection in standard deviation caught at approximatley 13 $w_b$ indicates that the diversity metric is more consistent between the different samples used for the iteration, suggesting that each individual sample is more likely to be representative of the natural variability in the channel. Beyond this scale, additional variability is captured, but at a decreasing rate. The 3.0 km of channel over which this analysis was conducted would likely be considered a relatively homogeneous riffle-pool reach under traditional channel classification schemes, such as that of Montgomery and Buffington (1997). The 15 $w_b$ length scale

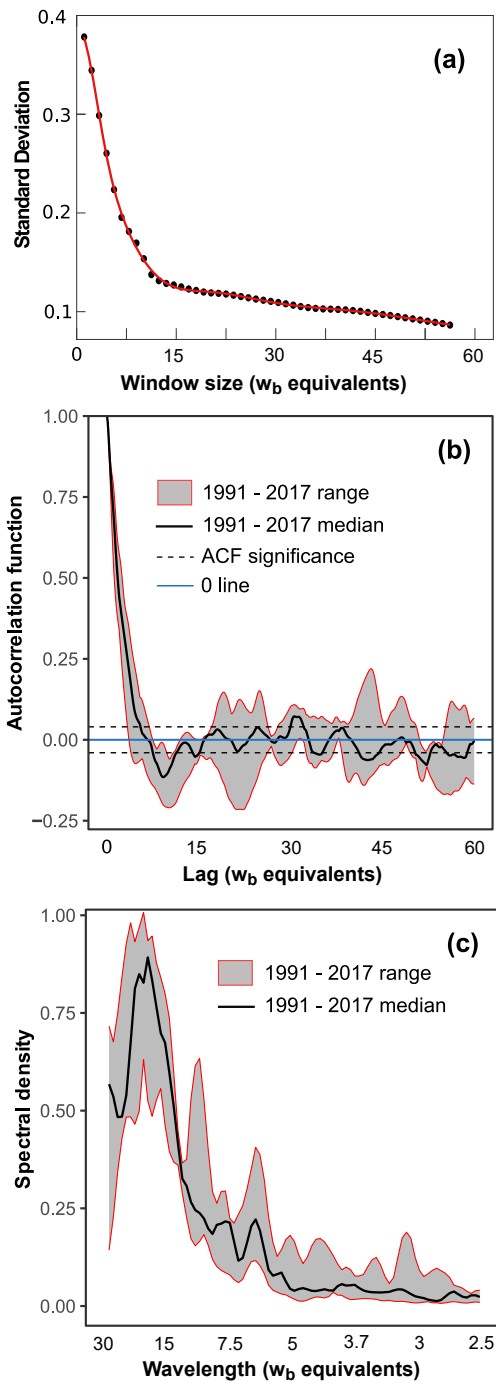

**Figure 10.** Notable length scales along the lower 3.0 km of Carnation Creek: (a) Standard deviation of channel diversity index values; (b) Autocorrelation function values extracted from channel longitudinal profile data collected four times between 1991 and 2017 (Fig. modified from Reid et al. (2019)); (c) Spectral density plot from analysis applied to longitudinal profile data in (b) (Fig. modified from Reid et al. (2019)). Note that channel width equivalents are given in relation to width determined as of 2017, equivalent to 13.4 m

is shorter than the 30–50 $w_b$ equivalent often suggested for characterizing channel form (Bisson et al., 2006), and equivalent to 2–3 sets of pool-riffle units as defined by Keller and Melhorn (1978). This value fits in with the range of recommended study reach lengths that have been reported in the literature, though it is at the lower end (see Trainor and Church, 2003). For example, Montgomery and Buffington (1997) considered reaches 10–20 $w_b$ in length for their research while Woodsmith and Buffington (1996) considered reaches 20 $w_b$ in length. At the higher end, Hogan (1986) and Trainor and Church (2003) consider reaches greater than 30 $w_b$ and reaches between 50–70 $w_b$ to be conservative lengths for their research, respectively. Given that additional variability is still captured with a greater spatial survey extent, the 15 $w_b$ value should be considered a minimum.

The explanation for the 15 $w_b$ domain over which a threshold in variability is reached may be related to the spacing of major sediment storage areas in the system. Previous work in Carnation Creek by Reid et al. (2019) suggests that non-random spatial patterns in sediment storage are present along the channel (see Fig. 10 b and c). Both autocorrelation and spectral analysis methods applied to four sediment storage datasets collected between 1991 and 2017 revealed a periodicity in the data in the order of 12-20 $w_b$, providing information on the spacing of major sediment storage areas. Given the similarity in length scales between Figs. 10 a-c, it is possible that these storage zones (mainly large bars) serve as end members between which the typical progression of channel unit morphologies would be expected.

The bar-to-bar spacing represented by length scales shown in Fig. 10 is within the range, but close to the upper limit, of values reported for gravel bed streams in Thompson (2013). The explanation for the relatively large feature spacing may be related to the presence of major logjams along the channel, which are commonly associated with areas of major sediment storage (Abbe and Montgomery, 1996; Davidson and Eaton, 2015; Wohl and Scott, 2017). However, as of 2017 (one year prior to the RPA survey) comparatively few major jams storing large quantities of sediment remained in the channel, and average jam spacing was only between 5 and 8 $w_b$ (see Reid et al., 2019). Other factors which may explain the relatively large unit spacing in Carnation Creek could be related to patterns in channel width (Chartrand et al., 2018) or flow convergence (MacVicar and Roy, 2007; Thompson and Wohl, 2009).

It is important to note that the spatial scale of measurement needed to capture variability will depend on the particular variables of interest, and also the expected morphological character of the system. Carnation Creek is a channel which experiences episodic delivery of sediment from hillslopes (Hartman and Scrivener, 1990; Reid et al., 2019). As shown by the range of values in Fig. 10 b and c, temporal variability exists in the spatial pattern of dominant channel features. The 26 year period over which the data in Fig. 10 b and c were collected represents a comparatively inactive time interval in terms of colluvial sediment supply. This variability would be expected to increase during periods of episodic sediment supply, and could influence the resulting spatial scale over which dominant variance is captured. In this instance, a greater length of channel may be necessary to survey in order to increase the probability of capturing this type of localized feature. Similarly, practical survey limitations (such as site accessibility) may still factor strongly in decisions regarding site selection and survey extent. As others (e.g. Montgomery and Buffington, 1998) have suggested, examination of channel gradient or a channel profile will still provide useful preliminary information on regions of relatively homogeneous channel morphology.

## 6 Conclusions

The spatial extent needed to adequately capture variability and classify morphology of forested, gravel bed streams with closed canopies is often unclear, while the challenge of collecting comprehensive data in these environments necessitates efficient and low-cost data acquisition methods. This paper describes an approach to characterise and classify these channels through use of sub-canopy flights with Remotely Piloted Aircraft (RPA) at the channel unit to reach scale. Through the incorporation of oblique-convergent imagery, it was possible to undertake a sub-canopy channel survey along 3.0 km of Carnation Creek, a small forested gravel-bed stream. Use of RPA-derived rasters of bed morphology, bathymetry, and grain size in combination with a PCA-clustering analysis of channel unit morphologies provided characterization of this channel at an extent and resolution that would be difficult to attain using traditional methods. This allowed for the exploration of the spatial extent necessary to capture the dominant morphological variability of the channel. After calculating a diversity index describing the heterogeneity in channel unit morphology, a spatial scale equivalent to approximately 15 channel widths was found to capture much of the variability in channel unit morphology.

Overall, the methods were successful in demonstrating the use of RPAs for collecting channel attribute data below forest canopies and in providing an objective technique for characterizing patterns in morphological units of small, forested channels at a variety of spatial scales. This research helps to expand the toolkit available to geomorphologists for characterizing small channels with complex morphology residing largely below forest canopies, and presents a classification approach with fewer drawbacks from subjective morphology identification. The results of this work are presented for a single catchment; additional study is needed to evaluate the limits of RPA approaches for data collection in similar environments.

*Data availability.* Data used for the analysis can be found at doi: 10.17632/jv9rftdmst.1 (Helm, 2020).

*Author contributions.* CH led all data collection, analysis and most manuscript preparation. MH provided supervisory support and assisted with project conceptualization and manuscript preparation. DR assisted with project conceptualization, data collection, and manuscript preparation.

*Competing interests.* The authors declare that they have no conflict of interest.

*Acknowledgements.* The fieldwork completed at Carnation Creek was made possible through a number of people involved in the watershed study. Robin Pike and Peter Tschaplinski supplied the reference data used in the project and helped coordinate the field work. Steve Voller and Andrew Westerhof provided much-appreciated support at the field site. Many thanks to Stephen Bird and John Richardson for their insightful discussions and feedback throughout the project. Ryan Matheson and Kyle Wlodarczyk provided field assistance for the data collected at

Carnation Creek. Eric Leinberger provided support in designing the figures. Jack Carrigan and Charles Helm proof-read the manuscript. The research was funded by NSERC Discovery (to M. Hassan) and Canada Foundation for Innovation (to M. Hassan).

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
