# Peer review of "Characterization of morphological units in a small, forested stream using close-range remotely piloted aircraft imagery"

_Earth Surface Dynamics, 2020_

## Referee Comment (RC1) · Anonymous Referee #1 · 1 Jun 2020

General comments:

The topic of the paper is interesting and tackles an important question related to the efficient field measurements of the river systems, which are having forest canopy. The paper is overall good, and especially the researchers from the fields of remote sensing and fluvial geomorphology will be interested in reading it. The authors have done huge work in field and with data processing. The methods are up-to-date and the paper is unique. However, before being possible to publish it, the manuscript would need clarifications in many sections, and rearrangement of the sentences / paragraphs. The terminology related to the spatial scales would be needed to define more precisely, so that readers would understand more easily what is meant with large, small etc. Overall, precision in the statements would make the paper more easily readable. The justifica-

tion of the paper would be needed to write more clearly in the introduction section. The texts and figures presented in the results and discussion sections would need also re-arranmegent. Also attention should be paid to the sub-titles. Overall, clarification of the text and justification of the importance of the selected topic, methods and gained results would be needed throughout the paper. Therefore, major modifications are suggested.

Specific comments:

Title of the paper: Consider deleting words rapid and objective from the title. Introduction or aims do not include these words, and justification and need of the rapidness of the techniques does not come clearly evident from the introducing sections. Or, if wanting to keep those words, add description about the rapidness and objectivity of the approach in the introduction section. I also suggest that the close-range remote sensing approach could be good to appear in the title some way or another.

Abstract: The following sentence is slightly contradictory, as you talk about both large areas and small streams. "This paper seeks to demonstrate an objective method for characterizing channel attributes over large areas, using easily extractable data from RPA imagery collected under the forest canopy in a small (width = 10 to 15 m) stream.." What do you mean with large areas? Could you clarify and modify the sentence so that it does not cause the reader to be confused between the different spatial scales under question.

Abstract: "The results demonstrate that sub-canopy RPA surveys provide a viable alternative to traditional survey approaches for characterizing these systems, with 87% coverage of the main channel stream bed." Cold you specify already here, what are the traditional survey approaches? Does this relate to the flight altitude? In addition, it would be actually important to also mention the flight altitudes (etc. details, which show how your method differed from the traditional approaches) in the abstract, as I would imagine that in the sub-canopy flights the height of the platform was low.

Lines 20-22: You mention that "These characteristics can lead to a high degree of spatial variability and . . .". Could you clarify the sentence, especially "spatial variability" of what? Both the first and second sentence of the introduction are slightly vague, and would need clarification, so that the start of the introduction would be stronger. It feels like there is repetition also in those first two sentences. Thus, make the beginning of the introduction sharper. In addition, it would be good to mention already in the first paragraph in detail what are the channel characteristics, which are important for the "management", and for the study, and why those are important? Is it only gradient, as that is the only one mentioned? The justification for the variables/metrics and their wider applicability does not come clear from the introduction. Therefore, the sharper beginning of the introduction and also more clearer justification for the study (parameters, and why their detection is important) would enable the reader to understand the uniqueness and importance of the paper more clearly.

Lines 50-54: The authors refer to Kasvi et al. (2019). That study has been done in a river system, having small channel width especially during the low flow periods. Therefore, please clarify the sentences so that the readers do not get an idea that Kasvi et al. (2019) paper has been done in larger river system. Again, please, define also in those lines 50-54, what do you mean with larger system / how do you define larger system?

Line 57: What is meant with "continuous RPA-derived data"? Is that spatially or temporally continuous?

Lines 100-101: What is the altitude of the low-level flights? Please, specify already here (i.e. where you first time mention these flying specifications), and not in he later sentences.

Lines 100-102: You write "The RPA survey involved low-level flights conducted in tandem with placement of Ground Control Points (GCPs) that were surveyed with a Leica TPS 1100 total station." Did you take the reference points from the sub-water

**ESurfD**
[Figure]

areas also? Or how do you calculate the accuracy of the bathymetry cells, which you talk about in results section 4.1? Please add in the methods section clearly, how the reference points for these RMSE and ME calculations were measured, and did you measure them also from the sub-water area and how (also with a total station similarly as the dry land areas and the GCPs)? Thus, some clarification and sharpness would be needed to the methods section also.

Line 114: You mention riparian vegetation here for the first time. How high is the riparian vegetation and what are the species. Was there grass and shrubs, or do you mean the "dense forest canopy composed of both coniferous and deciduous tree species", which you talk about in the study site section? In addition to mentioning the heights of the riparian vegetation (which were cleaned away from the data based of the filters), it would be good to also introduce the riparian vegetation in the study site section.

Relates to the methods and discussion section: Did the canopy effect on the pixel values of the water area? As you defined the bathymetry based on Dietrich et al. (2017) method, did the shadows and reflections of the canopy harm the water pixel colors and bathymetry calculations? What was the turbidity of the water? That information would be important to add, from the measurement times. The success of the Dietrich et al. (2017) method could depend on how turbid / clear the water was. Please, discuss about this in the discussion, and present how the turbidity was taken into account in the methods section.

Lines 139 -145: The authors introduce here the method for grain size estimation. However, this is the first time grain sizes are mentioned in the manuscript. Thus, there is no background literature in the introduction section, or justification why this calculation is important to conduct. You mention "a metric often of interest to river managers", but it would be important to justify here, why these metrics are important for your study. Please, add in the introduction and/or in the methods section, why the grain size is needed to be defined. To some readers the necessity to define the grain sizes is not

self-evident.

Line 173: I am not a native English speaker, but I think this following part of the sentence is missing one preposition "data along the fi̧rst" -> change it as "data along WITH the first".

Many of the figures appear only within the discussion section, and the results defined in some of the figures are not analyzed in detail in the text of the results section. For example, Fig. 8 appears on page 15, but it is talked with two sentences on the page 10. Thus, rearrange the appearance of the figures so that the text and figures appear "hand-in-hand". Despite the channel morphology was one of the main topics talked in the introduction section, the channel variables and the results of the morphological detection have not been given full attention in the results section. So, please, add text in the results section related to the morphological characteristics and their spatial variation.

Discussion: Many of the sentences (such as on lines 249- 255, and 278–285) should already be presented in the results section. Therefore, rearrangement of the discussion would be needed. I am not pointing out all of the sentences in question, as there are many of them. My advice is that when you present something for the first time based on your analysis or the data sets, move those sentences under results section. Discussion is then reflection of your results (presented already previous sections) against other studies.

Many of the sub-titles of the results and discussion section are methodological in their nature. Go through the titles of the manuscript and modify them so that they show that it is results and discussion in question, and not an introduction to the methods. Now the titles give slightly different idea of the content than what the content actually is: such as, "5.2 Classification approach" sounds like the section would include an explanation how the classification method was used, even though it is discussed about the "success of the classification approach". Thus, the titles of the results and discussions sections are

misleading.

**ESurfD**

---

## Referee Comment (RC2) · Anonymous Referee #2 · 3 Jul 2020

General Comments:

The paper presents a novel and useful methodology for mapping channel morphology that is well within the scope of ESurf. The methods were sound, logical and well presented. The introduction and discussion for the paper could use some adjustments, in particular clarification of the use of terminology such as channel morphology, morphological units, channel units, channel type, and morphology type. It was difficult to follow what was meant by each of these terms and if they were being used interchangeably or not. From the introduction I was expecting more of a reach scale channel type classification scheme, but I would argue that what this paper does would be better described as mapping or classification of morphological units (also called geomorphic units, channel units, habitat units, etc).

[Figure]

Specific Comments:

Title: I recommend the title including that the method uses a RPA or remote sensing

Abstract: Line 6 states "This paper seeks to demonstrate an objective method for characterizing channel attributes over large areas, using easily extractable data from RPA imagery collected under the forest canopy in a small stream, and to provide information on the spatial scale necessary to cacpture the dominant spatial morphological variability of these channels." - Rather than saying "characterizing channel attributes" it would be more precise for the author to say they are classifying or mapping channel morphological units. - provide clarification to what constitutes "large areas" - in "provide information on the spatial scale" does spatial scale mean longitudinal spatial extent?

Abstract: Line 14 "for characterizing these systems" it also would be better here to be more precise about mapping or classifying morphological units

Introduction: paragraphs 1 and 2 were confusing and misleading to me and could use clarification between reach scale stream classification and smaller, geomorphic/morphological/channel unit scale.

Methods: Line 146 says that in-stream wood was digitized, but I did not see this used or relevant later in the paper

Section 3.3: The author states that the 5 variables were chosen in part "because they reflect larger basin scale variables relevant to channel form, such as geology, climate and land use." A citation and/or examples here seem necessary

Analysis: Section 3.4: line 178 describes how a morphology type is attributed to each cluster. It would be helpful to lay out prior to this what the morphology types being used are, and the criteria used for them. The author does cite 3 papers for the criteria, but it isn't clear what specific criteria from those papers were used. Also, within this paragraph it isn't clear if morphology type is synonymous with channel type or not.

Figure 6: needs a scale bar

Conclusion: The conclusion would be easier to follow if it were organized in the same order as the rest of the paper.

---

## Referee Comment (RC3) · Anonymous Referee #3 · 7 Jul 2020

This manuscript presents the results from an investigation to generate a high-resolution orthoimage / topographic survey of c. 3 km of channel that is beneath a forest canopy. The geospatial products are used to extract metrics to characterise channel morphology, which are subsequent used to characterise longitudinal variation in channel morphology and to assess these trends relative to those reported in wider literature. The survey effort is impressive and undoubtably novel in its ambition; I am not aware of a similar survey. However, there are aspects of the methodological description that are unclear. I also think that the authors are overselling the approach as rapid and are not sufficiently critical of how transferable the technique may be to other forested environments (e.g. where canopy densities vary, where launching the UAV from under the canopy may be more challenging, for forested rivers without unvegetated bars etc) and

thus a more critical analysis of the technique could be provided. Below, I expand upon these aspects.

Title. Is the technique really rapid? The survey effort is still considerable from both a UAV flight and ground control perspective, and there are still some data gaps where total station survey is needed. L9-11. More methodological detail could be provided here; reading the abstract alone I'm not able to decipher what exactly how the imagery were used (orthoimage, DEM etc). Can you give examples of the "variables". L101. More details are needed on the total station control network and associated errors (e.g. closure errors from traverses). How accurate are the control points? L104. How high was the canopy? Were the flight plans pre-planned or was the UAV operated manually? You are arguing that this is a feasible survey approach, so some more details on the logistical / technical challenges would be useful here. If you had a pre-programmed flight, then could you obtain sufficient GNSS "lock" for navigation? L135. Were any independent total station check points obtained to evaluate the accuracy of the bathymetric correction? L141. Insufficient detail is given on the photo-sieving technique and how images were acquired from 2 m above ground level. If topographic roughness was calculated then were the multiple images acquired to generate a point cloud using SfM? How was the point cloud georeferenced? What was the ground sampling distance to sample 0.0025 cm roughness (smallest sample on figure 3; this seems VERY small). Are the units correct here? Figure 5. It would be interesting to see elevation values for the raster. Why are bathymetry and elevation not mosaiced together (perhaps it is the legend labelling that makes this unclear)? For SA5, what explains the abrupt change in bathymetry value (shown as a vertical line) in approximately the middle of the reach?

I apologise for the delay in my review which has been caused by unexpected, urgent administrative workload related to the pandemic, which arose after I agreed to review this manuscript.

---

## Author Comment (AC1) · 27 Jul 2020

Dear Associate Editor, and Reviewer 1:

Authors have responded to all comments below in full, with Reviewer 1 comments shown in bold, and responses as regular text. Notable new additions to the paper include, a simplified title, and sharpening of sections at the request of Reviewer 1. See below for detailed replies to all queries from Reviewer 1.

**REVIEWER 1 COMMENTS**

[Figure]

==============
**General comments**
==============

**The topic of the paper is interesting and tackles an important question related
to the efficient field measurements of the river systems, which are having forest
canopy. The paper is overall good, and especially the researchers from the
fields of remote sensing and fluvial geomorphology will be interested in reading
it. The authors have done huge work in field and with data processing. The
methods are up-to-date and the paper is unique. However, before being possible
to publish it, the manuscript would need clarifications in many sections, and
rearrangement of the sentences / paragraphs. The terminology related to the
spatial scales would be needed to define more precisely, so that readers would
understand more easily what is meant with large, small etc. Overall, precision in
the statements would make the paper more easily readable. The justification of
the paper would be needed to write more clearly in the introduction section. The
texts and figures presented in the results and discussion sections would need
also re-arranmegent. Also attention should be paid to the sub-titles. Overall,
clarification of the text and justification of the importance of the selected topic,
methods and gained results would be needed throughout the paper. Therefore,
major modifications are suggested.**

The authors are grateful for the constructive comments and thoughtful suggestions
from Reviewer 1. Improvements have been made to the paper to add precision to our
statements. In addition, we have incorporated more text into the introduction clarifying
why we believe RPAs should be considered for surveying small forested channels and
added justification for the variables included in our study. We have also sharpened
several of the subtitles in the paper and reorganized some portions of the text at the

request of Reviewer 1. These changes are further described in the queries below.

==============
**Specific comments**
==============

**Title of the paper: Consider deleting words rapid and objective from the title. Introduction or aims do not include these words, and justification and need of the rapidness of the techniques does not come clearly evident from the introducing sections. Or, if wanting to keep those words, add description about the rapidness and objectivity of the approach in the introduction section. I also suggest that the close-range remote sensing approach could be good to appear in the title some way or another.**

The comment has been accepted. At the suggestion of Reviewer 1, the title has been changed to "Characterization of morphological units in a small, forested stream using close range RPA imagery".

**Abstract: The following sentence is slightly contradictory, as you talk about both large areas and small streams. "This paper seeks to demonstrate an objective method for characterizing channel attributes over large areas, using easily extractable data from RPA imagery collected under the forest canopy in a small (width = 10 to 15 m) stream.." What do you mean with large areas? Could you clarify and modify the sentence so that it does not cause the reader to be confused between the different spatial scales under question.**

The intention was to highlight that the survey was conducted in a small forested

stream, over a large section of the channel's longitudinal profile (3 km). At the suggestion of Reviewer 1, we have modified this sentence in the abstract of the revised text with the following text:

"This paper seeks to demonstrate an objective method for classifying channel morphological units in small, forested streams and to provide information on the spatial scale necessary to capture the dominant spatial morphological variability of these channels. This was achieved using easily extractable data from close-range RPA imagery collected under the forest canopy (flying height = 5 - 15 m above ground level) in a small (width = 10 - 15 m) stream along its 3 km of anadromous salmon-bearing channel."

**Abstract: "The results demonstrate that sub-canopy RPA surveys provide a viable alternative to traditional survey approaches for characterizing these systems, with 87% coverage of the main channel stream bed." Cold you specify already here, what are the traditional survey approaches? Does this relate to the flight altitude?**

We have adjusted this line to show that the intent was to refer to ground-based approaches (e.g. total station, automatic level) with the following text:

"The results demonstrate that sub-canopy RPA surveys provide a viable alternative to traditional ground-based survey approaches for mapping morphological units. . ."

These classification approaches are further described in the introduction of the revised text, which have historically been better suited and more widely applied to streams like Carnation Creek, with the following text:

[Figure]

"Traditionally, characterization and classification of channels through field surveys has required the use of a variety of GPS-based tools and linear-survey methods involving automatic levels, theodolites, and total-stations"

**In addition, it would be actually important to also mention the flight altitudes (etc. details, which show how your method differed from the traditional approaches) in the abstract, as I would imagine that in the sub-canopy flights the height of the platform was low.**

This is correct, the flying height of the RPA was quite low. This information has been added to the abstract of the revised text with:

"This was achieved using easily extractable data from close-range RPA imagery collected under the forest canopy (flying height = $5 - 15$ m above ground level) in a small (width = $10 - 15$ m) stream along its 3 km of anadromous salmon-bearing channel."

**Lines 20-22: You mention that "These characteristics can lead to a high degree of spatial variability and...". Could you clarify the sentence, especially "spatial variability" of what? Both the first and second sentence of the introduction are slightly vague, and would need clarification, so that the start of the introduction would be stronger. It feels like there is repetition also in those first two sentences. Thus, make the beginning of the introduction sharper.**

We have merged these sentences together with the sentence below to clarify:

"Their classification may be particularly important in forested, gravel bed streams, where episodic and transient geomorphological processes (Pryor et al., 2011; Wohl and Brian, 2015; Hassan et al., 2019), can lead to a high degree of channel complexity, even within a relatively homogeneous channel type (Madej, 1999; Nelson et al., 2010; Gartner et al., 2015)."

At the suggestion of Reviewer 2, the first two paragraphs of the introduction haven been reworked for clarity and terminology as well.

**In addition, it would be good to mention already in the first paragraph in detail what are the channel characteristics, which are important for the "management", and for the study, and why those are important? Is it only gradient, as that is the only one mentioned? The justification for the variables/metrics and their wider applicability does not come clear from the introduction. Therefore, the sharper beginning of the introduction and also more clearer justification for the study (parameters, and why their detection is important) would enable the reader to understand the uniqueness and importance of the paper more clearly.**

The authors agree with this suggestion. This has been incorporated with the following lines:

"RPAs are likely advantageous in these systems, as they may easily permit the extraction of a greater set of variables to aid in channel classification. These variables include features such as channel slope, water depth, and grain size characteristics, all of which reflect larger basin-scale controls on channel morphology (Buffington and Woodsmith, 2003). Channel slope is a key variable to consider, as it has been

shown that there is a general progression of channel morphologies from pool-riffle, plane-bed, and step-pool to cascade morphologies with increasing slope (Montgomery and Buffington, 1997). Water depth metrics are important for discriminating between pool environments and other shallow water environments. Finally, grain size is a key variable, as there tends to be a coarsening in bed material from glides and pools to riffles and runs. Acquiring this suite of variables is a difficult task, one that RPAs may be uniquely suited to."

**Lines 50-54: The authors refer to Kasvi et al. (2019). That study has been done in a river system, having small channel width especially during the low flow periods. Therefore, please clarify the sentences so that the readers do not get an idea that Kasvi et al. (2019) paper has been done in larger river system. Again, please, define also in those lines 50-54, what do you mean with larger system / how do you define larger system?**

The authors appreciate this comment from Reviewer 1. Relative to Carnation Creek, the authors would consider the stream investigated by Kasvi et al., (2019) to be large. We have added the definition of channel size from Hassan et al., (2005), to the following lines of the revised text:

"However, much of this work has been limited to larger systems. Herein we consider the classification by Hassan et al., (2005 ) for small to intermediate streams in the Pacific Northwest as those where the ratio between bankfull channel width to wood length is close to or greater than one and the ratio between log diameter to bankfull depth is close to or greater than one (see Table 2 of the paper for more details). Streams on the intermediate side of this spectrum, where the ratio between bankfull channel width to wood length is close to one, differ from larger systems as they can be greatly influenced by wood delivered to the channel. They are often overlain by dense

forest canopies that are poorly suited to observation from above the forest canopy. This limitation has historically excluded a large fraction of river network length from RPA-based surveys."

**Line 57: What is meant with "continuous RPA-derived data"? Is that spatially or temporally continuous?**

The authors had intended to be referencing the fact that data acquired from an RPA is continuous across the channel (i.e. space, rather than time). By contrast, field methods for surveying channels often involve discrete cross sections or points that must be interpolated. This has been clarified by adding "spatially continuous" to the sentence.

**Lines 100-101: What is the altitude of the low-level flights? Please, specify already here (i.e. where you first time mention these flying specifications), and not in he later sentences.**

We have rearranged the paragraph so that the flying specifications appear when the RPA survey is first introduced in the revised text.

**Lines 100-102: You write "The RPA survey involved low-level flights conducted in tandem with placement of Ground Control Points (GCPs) that were surveyed with a Leica TPS 1100 total station." Did you take the reference points from the sub-water areas also? Or how do you calculate the accuracy of the bathymetry cells, which you talk about in results section 4.1? Please add in the methods section clearly, how the reference points for these RMSE and ME calculations were measured, and did you measure them also from the sub-water area and**

**how (also with a total station similarly as the dry land areas and the GCPs)? Thus, some clarification and sharpness would be needed to the methods section also.**

The authors have reworded these lines to make the survey method clearer. To reiterate, all GCPs and checkpoints were surveyed with a Leica TPS 1100 total station. The GCPs were only positioned on the dry exposed bars, whereas checkpoints include both the dry exposed bars and submerged points. This is described in the following lines of the revised text:

"A minimum of ten ground control points (GCPs) were placed along dry exposed bars in each of the 80 channel segments to provide precise image georeferencing, with additional points positioned on the dry exposed bars and below the water surface in order to serve as independent checkpoints, to assess the accuracy of the model outputs. All GCPs and checkpoints were surveyed with a Leica TPS 1100 total station. The majority of the GCPs were distributed in a zig-zag fashion along dry exposed bars in the periphery of the channel segments, with a smaller number situated towards the centre."

**Line 114: You mention riparian vegetation here for the first time. How high is the riparian vegetation and what are the species. Was there grass and shrubs, or do you mean the "dense forest canopy composed of both coniferous and deciduous tree species", which you talk about in the study site section? In addition to mentioning the heights of the riparian vegetation (which were cleaned away from the data based of the filters), it would be good to also introduce the riparian vegetation in the study site section.**

This comment has been accepted and is addressed with the following lines in the

study area section of the revised text:

"The riparian vegetation includes a variety of tree species including western hemlock (Tsuga heterophylla), Amabilis fir (Abies amabilis), western redcedar (Tsuga plicata), Sitka spruce (Picea sitchensis) and red alder (Alnus rubra). The height of the riparian canopy is variable, between approximately 15 and 40 m. The riparian forest floor is composed of a variety of ferns and shrubs, e.g., salmonberry (Rubus spectabilis), sword fern (Polystichum munitum), trailing blackberry (Rubus ursinus) and thimble-berry (Rubus parviflorus), that may provide some cover to the channel."

**Relates to the methods and discussion section: Did the canopy effect on the pixel values of the water area? As you defined the bathymetry based on Dietrich et al. (2017) method, did the shadows and reflections of the canopy harm the water pixel colors and bathymetry calculations? What was the turbidity of the water? That information would be important to add, from the measurement times. The success of the Dietrich et al. (2017) method could depend on how turbid / clear the water was. Please, discuss about this in the discussion, and present how the turbidity was taken into account in the methods section.**

At the suggestion of Reviewer 1, we added additional text pertaining to this in the revised text with:

"The method requires that the water be clear such that the channel bed can be captured. The low flow conditions present at the time of the survey resulted in clear water that permitted viewing of the channel bed. Removal of overhanging vegetation using the Cloth Simulation Filter in Cloud Compare, and subsampling the DEMs to a spacing of 0.02 m using the minimum elevations in the point cloud, helped to

ensure that the refraction correction was based on channel bed points, and not on overhanging vegetation points that may have been incorporated in the point cloud."

As now described in the paper, Carnation Creek was very clear and turbidity was not an issue at the time of the survey. The point cloud was cleaned to remove anomalous points from overhanging vegetation that may have been incorporated into the cloud. The individual effect of factors such as shadows and reflections on the bathymetry calculations was not investigated, but rather the total errors between RPA derived submerged elevations and total station measured elevations presented in Fig. 4.

**Lines 139 -145: The authors introduce here the method for grain size estimation. However, this is the first time grain sizes are mentioned in the manuscript. Thus, there is no background literature in the introduction section, or justification why this calculation is important to conduct. You mention "a metric often of interest to river managers", but it would be important to justify here, why these metrics are important for your study. Please, add in the introduction and/or in the methods section, why the grain size is needed to be defined. To some readers the necessity to define the grain sizes is not self evident.**

See our reply to your previous comment on "In addition, it would be good to mention already in the first paragraph in detail what are the channel characteristics, which are important for the "management", and for the study, and why those are important? ". We have reiterated this in the methods section where we note that grain size is a frequently described metric in classification schemes, such as Montgomery and Buffington's (1997) classification scheme.

**Line 173: I am not a native English speaker, but I think this following part of the**

[Figure]

**sentence is missing one preposition "data along the**... **change it as "data along WITH the first".**

We have clarified this sentence with:

"Following the PCA, the k-means clustering algorithm was run to identify groupings that may have been present in the data along its first three components."

**Many of the figures appear only within the discussion section, and the results defined in some of the figures are not analyzed in detail in the text of the results section. For example, Fig. 8 appears on page 15, but it is talked with two sentences on the page 10. Thus, rearrange the appearance of the figures so that the text and figures appear "hand-in-hand".**

The authors accept this comment. The figures have been rearranged such that the text and figures appear "hand-in-hand".

**Despite the channel morphology was one of the main topics talked in the introduction section, the channel variables and the results of the morphological detection have not been given full attention in the results section. So, please, add text in the results section related to the morphological characteristics and their spatial variation.**

The authors accept this comment. We have added text to section 4.2.1 further describing differences between the identified channel types and their relative positions:

"However, plane-bed and coarse riffle morphological units are mostly located near the upstream limit of the survey extent in this region. This area represents the outlet and downstream entrance of the canyon reach, where steeper gradients and coarser sediment are found. This is highlighted in Table 2, which shows that on average these morphological units are located 3160 m upstream, with steep reach scale slopes of 0.042 m/m and coarse material with an average D50 of 8.21 cm. Similarly, the coarse riffle morphologies were located approximately 2980 m upstream on average, with relatively steep slopes and coarse material (reach scale slope = 0.0024 m/m and D50 = 6.74 cm). By contrast, the average positions of the riffle, glide, run and pool morphologies were approximately 1500 m, midway along the channel's profile, which follows suit with the uniform distribution of these morphological units. Grain size was generally similar between these morphological units, except for the riffle unit, which was slightly coarser with a D50 of 4.10 cm. Pools were the deepest, with average water depths of 1.04 m and near zero water surface slopes, whereas riffles were the shallowest with average water depths of 0.13 m and relatively steep water surface and reach scale bed slopes. Glide and runs were intermediate between these morphologies, with glides often retaining negative local slopes, corresponding with the exit of pools, and runs with large positive local slopes, corresponding with the entry of pools.

**Discussion: Many of the sentences (such as on lines 249- 255, and 278–285) should already be presented in the results section. Therefore, rearrangement of the discussion would be needed. I am not pointing out all of the sentences in question, as there are many of them. My advice is that when you present something for the first time based on your analysis or the data sets, move those sentences under results section. Discussion is then reflection of your results (presented already previous sections) against other studies.**

The authors appreciate this perspective from the reviewer. It was the intention of the authors to have the Discussion reiterate the key findings of the study, and then introduce relevant references that help situate these findings. We have reworded lines 249-255 of the original text to make relationships between reiterated findings and references included clearer.

Regarding lines 278-285 of the original text, the table those lines discuss was included to make it easier for the reader to compare the mean values of the morphologies in our study to others in the literature. The values presented in the table for our study were extracted from Table 2 in the results. As suggested by Reviewer 2, we have also added a new Table 1 to the introduction summarizing the criteria for the classification schemes by Church (1992), Anonymous (1996) and Buffington and Woodsmith (2003). As those lines situate the results of our classification scheme in the literature, the authors feel they are still suited to the discussion and are in line with the reviewers suggestion that the "Discussion be a reflection of your results (presented already previous sections) against other studies".

**Many of the sub-titles of the results and discussion section are methodological in their nature. Go through the titles of the manuscript and modify them so that they show that it is results and discussion in question, and not an introduction to the methods. Now the titles give slightly different idea of the content than what the content actually is: such as, "5.2 Classification approach" sounds like the section would include an explanation how the classification method was used, even though it is discussed about the "success of the classification approach". Thus, the titles of the results and discussions sections are misleading.**

The authors accept this comment. As suggested by Reviewer 1, we have changed the titles for the following sections: 4.1, 4.2. 4.2.1, 4.3, 5.1, 5.2.

**ESurfD**

Interactive
comment

---

## Author Comment (AC2) · 28 Jul 2020

Dear Associate Editor, and Reviewer 2:

Authors have responded to all comments below in full, with Reviewer 2 comments shown in bold, and responses as regular text. Notable new additions to the paper include implementing a consistent terminology for the morphological units the manuscript describes mapping. Please see below for detailed replies to all queries from Reviewer 2.

**REVIEWER 2 COMMENTS**

[Figure]

==============
**General comments**
==============

**The paper presents a novel and useful methodology for mapping channel morphology that is well within the scope of ESurf. The methods were sound, logical and well presented. The introduction and discussion for the paper could use some adjustments, in particular clarification of the use of terminology such as channel morphology, morphological units, channel units, channel type, and morphology type. It was difficult to follow what was meant by each of these terms and if they were being used interchangeably or not. From the introduction I was expecting more of a reach scale channel type classification scheme, but I would argue that what this paper does would be better described as mapping or classification of morphological units (also called geomorphic units, channel units, habitat units, etc).**

The authors are grateful for the constructive comments from Reviewer 2. In particular, the comment on using a consistent channel morphology terminology is an important one. The authors have updated the manuscript such that we are consistent with referring to the scheme classifying "morphological units" or "channel units". Please see below for detailed responses to further queries.

==============
**Specific comments**
==============

**Title: I recommend the title including that the method uses a RPA or remote sensing.**

The comment has been accepted. At the suggestions of Reviewer 1, Reviewer 2 and Reviewer 3, the title has been changed to "Characterization of morphological units in a small, forested stream using close range RPA imagery".

**Abstract: Line 6 states "This paper seeks to demonstrate an objective method for characterizing channel attributes over large areas, using easily extractable data from RPA imagery collected under the forest canopy in a small stream, and to provide information on the spatial scale necessary to cacpture the dominant spatial morphological variability of these channels." - Rather than saying "characterizing channel attributes" it would be more precise for the author to say they are classifying or mapping channel morphological units. - provide clarification to what constitutes "large areas" - in "provide information on the spatial scale" does spatial scale mean longitudinal spatial extent?**

This comment has been accepted. At the suggestion of Reviewer 2, we have replaced "characterizing channel attributes" with "classifying channel morphological units". Furthermore, at the suggestions of Reviewers 1 and 2, we have clarified the spatial scales under investigation with the text below:

"This paper seeks to demonstrate an objective method for classifying channel morphological units in small, forested streams and to provide information on the spatial scale necessary to capture the dominant spatial morphological variability of these channels. This objective was achieved using easily extractable data from close-range RPA imagery collected under the forest canopy (flying height = $5 - 15$ m above

ground level) in a small (width = 10 – 15 m) stream along its 3 km of anadromous salmon-bearing channel."

**Abstract: Line 14 "for characterizing these systems" it also would be better here to be more precise about mapping or classifying morphological units.**

This comment has been accepted. At the suggestion of Reviewer 2, we have replaced "for characterizing these systems" with "mapping morphological units".

**Introduction: paragraphs 1 and 2 were confusing and misleading to me and could use clarification between reach scale stream classification and smaller, geomorphic/morphological/channel unit scale.**

This comment has been accepted. At the suggestion of Reviewer 1 and Reviewer 2, the first two paragraphs of the introduction have been reworded for clarity and to describe that the analysis is aimed at classifying morphological units:

"Channel morphological units such as pools and riffles constitute the building blocks for reach scale channel morphologies (Buffington and Montgomery, 2013a). Variability in these units within a channel reach can provide critical habitat diversity. As a result, characterization of morphological units is the goal of many habitat-based classification schemes (e.g. Hawkins et al., 1993). Their classification may be particularly important in forested, gravel bed streams, where episodic and transient geomorphological processes (Pryor et al., 2011; Wohl and Brian, 2015; Hassan et al., 2019), can lead to a high degree of channel complexity even within a relatively homogeneous channel type (Madej, 1999; Nelson et al., 2010; Gartner et al., 2015). For these streams, classification schemes can serve an important role in facilitating

discussions on stream management among disciplines (Buffington and Montgomery, 2013b). This is evident in the array of classification schemes proposed to characterize channel types and morphological units for both geomorphologists and ecologists alike (e.g. Hawkins et al., 1993; Rosgen, 1994; Montgomery and Buffington, 1997; Brierly and Fryirs, 2005). A common challenge of these classification approaches, however, is their descriptive nature (Buffington and Montgomery, 2013b; Hassan et al., 2017) and that their implementation can be subjective, differing between classifiers.

Challenges in objectively classifying morphological units are further compounded by difficulties in determining the appropriate spatial extent for capturing the primary structural variability that influences geomorphological and ecological processes at the reach or basin scale. While approaches are often taken to select 'representative sites' when the characterisation of channel variables is necessary (Harrelson et al., 1994; Bisson et al., 2006), site selection is often based on a narrow subset of metrics (e.g. gradient, see Montgomery and Buffington, 1998) and 'rules of thumb' are frequently used to define the spatial extent of the surveyed area (Bisson et al., 2006). Furthermore, traditional survey techniques often limit classification to short, accessible channel areas due to time and cost constraints, and these limitations may bias our understanding of the larger river network as a result of missing important channel areas and processes (Fausch et al., 2002; Hugue et al., 2016). Given the logistical difficulty and cost of undertaking field surveys in small, forested gravel-bed streams, a more precise approach for site selection and objective technique for classifying morphological units is warranted."

**Methods: Line 146 says that in-stream wood was digitized, but I did not see this used or relevant later in the paper**

The authors accept this comment. Channel wood was digitized as part of the initial

inventory of channel variables but was not used in the analysis. The line has been removed from the revised manuscript.

**Section 3.3: The author states that the 5 variables were chosen in part "because they reflect larger basin scale variables relevant to channel form, such as geology, climate and land use." A citation and/or examples here seem necessary**

The authors have cited Buffington and Woodsmith, 2003 here. Specifically Figure 1 of their paper shows how the channel characteristics we included give rise to channel types, and are the manifestation of certain process drivers and watershed conditions. In addition, at the suggestion of Reviewer 1, we have added the following text explaining why those variables were selected:

"Channel slope is a key variable to consider, as it has been shown that there is a general progression of channel morphologies from pool-riffle, plane-bed, and step-pool to cascade morphologies with increasing slope (Montgomery and Buffington, 1997). Water depth metrics are important for discriminating between pool environments and other shallow water environments. Finally, grain size is a key variable, as there tends to be a coarsening in bed material from glides and pools to riffles and runs (Garcia et al., 2012)."

**Analysis: Section 3.4: line 178 describes how a morphology type is attributed to each cluster. It would be helpful to lay out prior to this what the morphology types being used are, and the criteria used for them. The author does cite 3 papers for the criteria, but it isn't clear what specific criteria from those papers were used. Also, within this paragraph it isn't clear if morphology type is synonymous with channel type or not.**

**ESurfD**
The authors accept this comment. The morphological units being classified have now been introduced prior to introducing the criteria. A table (now Table 1) has also been added summarizing the criteria extracted from these papers. The terminology has also been changed to morphological unit to ensure consistency through the manuscript:

"Following clustering of the cross-sectional variables, the mean values of each channel variable for each cluster were examined and one of the following morphological units attributed to each cluster: pool, riffle, coarse riffle (riffleC), glide, run or plane-bed. The units were assigned to clusters based on obvious features (e.g. shallow water slopes and greater depth for pools, negative pool exit slopes for glides, and steeper pool entry slopes for runs) and criteria presented in Church (1992), Anonymous (1996), and Buffington and Woodsmith (2003). These criteria are described in Table 1. The resulting assignment of morphologies to clusters leads to a continuous classification of morphological units found along the study reach at 1 m intervals, and provides insight into the survey extents necessary to adequately capture the heterogeneity of the system."

**Figure 6: needs a scale bar**

This authors accept this comment. A scale bar has been added to Figure 6.

**Conclusion: The conclusion would be easier to follow if it were organized in the same order as the rest of the paper.**

The authors accept this comment. We have rearranged the paragraph so that the acquisition of RPA derived rasters and ensuing PCA comes first, followed by the sentence on the exploration of the necessary spatial extent to capture the channels variability. This now follows the order of the methods.

---

## Author Comment (AC3) · 28 Jul 2020

Dear Associate Editor, and Reviewer 3:

Authors have responded to all comments below in full, with Reviewer 3 comments shown in bold, and responses as regular text. Notable new additions to the paper include sharpening of several methodological components of the manuscript.

**REVIEWER 3 COMMENTS**

==============

**General comments**
==============

This manuscript presents the results from an investigation to generate a high-resolution orthoimage / topographic survey of c. 3 km of channel that is beneath a forest canopy. The geospatial products are used to extract metrics to characterise channel morphology, which are subsequent used to characterise longitudinal variation in channel morphology and to assess these trends relative to those reported in wider literature. The survey effort is impressive and undoubtably novel in its ambition; I am not aware of a similar survey. However, there are aspects of the methodological description that are unclear. I also think that the authors are overselling the approach as rapid and are not sufficiently critical of how transferable the technique may be to other forested environments (e.g. where canopy densities vary, where launching the UAV from under the canopy may be more challenging, for forested rivers without unvegetated bars etc) and thus a more critical analysis of the technique could be provided. Below, I expand upon these aspects.

The authors are grateful for the constructive comments and detailed review from Reviewer 3. The authors have included text to sharpen several aspects of the methodology. Please see below for detailed responses to further queries.

==============
**Specific comments**
==============

**Title. Is the technique really rapid? The survey effort is still considerable from**

**both a UAV flight and ground control perspective, and there are still some data gaps where total station survey is needed.**

Relative to a dense total station survey, the authors considered the acquisition of continuous data to be rapid. As noted in the Discussion, with the RPA we could capture, on daily average, three times more of the channel's length than that covered with a total station full channel survey. However, relative to other approaches (such as an automatic level survey of the channels longitudinal profile, or RPA surveys done at a greater height above ground level), it may not be considered rapid. As this detail cannot be discerned from a short title, the authors accept the comment and have removed "Rapid" from the title for:

Characterization of morphological units in a small, forested stream using close-range RPA imagery.

**L9-11. More methodological detail could be provided here; reading the abstract alone I'm not able to decipher what exactly how the imagery were used (orthoimage, DEM etc). Can you give examples of the "variables".**

The authors accept this comment, and have incorporated the following text into the abstract:

"From this survey data, relevant cross-sectional variables (hydraulic radius, sediment texture and channel slopes) were extracted from high resolution point clouds and DEMs of the channel, and used to characterize channel unit morphology using a principal component analysis-clustering (PCA-clustering) technique."

**L101. More details are needed on the total station control network and associated errors (e.g. closure errors from traverses). How accurate are the control points?**

The total station survey was an open survey, as it was not feasible to re-survey the 3 km of channel to the close the traverse. Rather, the open traverses were tied into the benchmarks previously established in the study sections. This has been clarified with the following lines:

"Open survey traverses were tied into the benchmarks previously established in the study sections, and then an affine transformation applied to georeference the points in the XY-plane. The average offset between the benchmark elevations of the local open traverse and their known reference elevations were then used to georeference the points in the Z-plane. Errors were typically 2 cm in the XY-plane, and 1 cm in the Z-plane."

**L101L104. How high was the canopy?**

At the suggestion of Reviewer 1, we have added more details on the forest characteristics surrounding the channel in the study area section:

"The height of the riparian canopy is variable, between approximately 15 and 40 m."

**Were the flight plans pre-planned or was the UAV operated manually? You are arguing that this is a feasible survey approach, so some more details on the logistical / technical challenges would be useful here. If you had a pre-**

**programmed flight, then could you obtain sufficient GNSS "lock" for navigation?**

The flights were conducted manually as due to channel obstacles, and the channel being hidden below the canopy, pre-planning the flights would have been challenging. We have added more text highlighting this with the following lines in section 3.1:

"The RPA survey involved low-level flights (5–15 m above ground level) conducted in tandem with placement of ground control points (GCPs) on the dry exposed bars and checkpoints on both the exposed and submerged bed. Flights were operated manually as channel obstacles such as overhanging vegetation, and the fact that much of the survey was conducted below the canopy, would make pre-planning flights a challenge"

**L135. Were any independent total station check points obtained to evaluate the accuracy of the bathymetric correction?**

Yes independent total station check points were obtained for the evaluating the bathymetric correction. See our response to the previous comment where we have clarified that we did have submerged checkpoints. In addition, we have added the following text:

"A minimum of ten GCPs were placed along dry exposed bars in each of the 80 channel segments to provide precise image georeferencing, with additional points positioned on the dry exposed bars and below the water surface in order to serve as independent checkpoints, to assess the accuracy of the model outputs. All GCPs and checkpoints were surveyed with a Leica TPS 1100 total station."

The errors are further highlighted in Fig. 4.

**L141. Insufficient detail is given on the photo-sieving technique and how images
were acquired from 2 m above ground level. If topographic roughness was
calculated then were the multiple images acquired to generate a point cloud
using SfM? How was the point cloud georeferenced? What was the ground
sampling distance to sample 0.0025 cm roughness (smallest sample on figure
3; this seems VERY small). Are the units correct here?**

The authors accept this comment. We have clarified how the sample sites were
photoseived, and how the point clouds were extracted, with the lines below:

"Each roughness sampling site was approximately 1 m2 and imagery was captured
for photo-sieving by hovering the RPA approximately 2 m above ground level. Us-
ing an in-house photo-sieving program based in Matlab (Matlab, 2017), the grain
size distributions of each training site were determined. The program loads the
image, prompts the user to scale the image, and then overlays a grid with 50
nodes prompting the user to measure the B-axis of grains falling below a grid node.
Point clouds for each sample site were then extracted from the georeferenced point
cloud that was developed for the study section that they fell within. A linear model
then was then fit between each sample's D50 (Fig. 3) and their mean roughness value."

The authors thank the reviewer for noticing the error in the x-axis units for Fig 3.
Initially the units were in m, but the figure has been updated so that they are in cm.

**Figure 5. It would be interesting to see elevation values for the raster. Why
are bathymetry and elevation not mosaiced together (perhaps it is the legend
labelling that makes this unclear)? For SA5, what explains the abrupt change in**

**bathymetry value (shown as a vertical line) in approximately the middle of the
reach?**

The intention of Fig. 5 was to show the relative coverage between the RPA survey
and total station surveys. Elevations vary from around sea level at the mouth of
Carnation Creek to about 140 m.a.s.l at SA9. This wide range in elevations would
require each SA have its own colour bar to effectively be able to visualize patterns in
elevation. Therefore, the terrain of the bed was instead shown as a hillshade layer
to show patterns in bed texture without resulting in a busy figure that may take away
from the Figure's purpose: to show the coverage with the RPA relative to the total
station surveyed boundaries. Instead, Figure 9 was included which shows patterns in
elevation along a section of the channel.

Fig. 4 in the manuscript shows that the majority of the approximatley 1700 checkpoint
errors were on the cm scale, indicating situations such as that observed in SA5 are
outliers in the larger survey effort. Visual inspection of the point cloud reveals that
this area, which is characterized as having dense low-lying vegetation on the fringe
of a new survey section, had a lower density of points, suggesting there may be
issues with the DEM produced. It should be noted that a strength of the PCA is that
it highlights patterns in a dataset, and not potentially noisy areas as in SA5. In other
words, although in the 3 km of channel there may be some anomalous values from
poorly aligned sections, these "outliers" should not affect the general trend identified
by the PCA . Text has been added to the manuscript highlighting these points in the
following sections.

In the "Utility of sub-canopy RPA surveys for small, forested streams" section, we have
clarified our sentence on image capture difficulties with:

"In addition to bank vegetation causing obstructions, submerged areas with poor texture and low-hanging branches (predominantly from riparian deciduous species) occasionally led to flight difficulties that prevented sufficient collection of imagery for photo-stitching."

In the "Assessment of classification approach" section, we added:

"Another advantage of the PCA is that it highlights the trends present in a dataset, rather than focussing on specific features. For example, anomalous areas where imagery may have had stitching issues due to poor coverage would likely appear as noise, thereby having a minimal influence on the PCA."

---

## Author Response (AR2)

*Dear Associate Editor:*

*Authors have responded to all comments below in full, with Associate Editor comments shown in bold, and responses as regular text. The authors greatly appreciate the insightful comments from the Associate Editor which will certainly help to enhance the clarity of the paper. See below for detailed replies to all queries.*

**ASSOCIATE EDITOR COMMENTS**

**################**
**General Comments**
**################**

**Thanks for your thorough and considered response to the reviewers' comments. I'm happy that you have addressed them all. I've been through the paper and identified a number of places where I think that minor edits would enhance the clarity of the paper, which you can see in the attached document. Please have a look through my suggestions and revise your paper accordingly.**

The authors are grateful for the constructive comments and thoughtful suggestions from the Associate Editor. All comments have been addressed to help enhance the clarity of the paper.

**################**
**Specific comments:**
**################**

**Title of the paper: RPA isn't an acronym that I was familiar with. If it's not that widely used, then spell it out in the title.**

Accepted, the title has been changed to "Characterization of morphological units in a small, forested stream using close range remotely piloted aircraft imagery".

**L50 – disjointed sentence "In consideration of a river network, these spatial scales are often intermediate in length (in the order of kilometers), domains over which continuous, high-resolution characterisation of channel conditions is expensive and time consuming using ground-based survey methods"**

The sentence has been reworded and additional text added to add clarity with:

"These spatial scales are often intermediate in length (in the order of kilometers), domains over which continuous, high-resolution characterisation of channel conditions has traditionally been a challenge due to the time and cost constraints of ground-based survey methods (Fausch et al., 2002). Over the past decade, the use of remotely piloted aircraft (RPAs) has helped overcome this challenge through the collection of high-resolution imagery over a range of scales for evaluation of stream bed topography…"

**L80 - Might be useful to very briefly explain what these data are. However, you don't really make that much use of this extra data, beyond that in Fig 10.**

The authors accept this comment. In the revised manuscript we have clarified that the complementary data used came from annual surveys of the channel's study sections, and a longitudinal profile survey along the channel's lower 3.0 km (see L84).

**L116 – "Not clear why smaller channel dimensions makes it more challenging - harder to fly between the trees?"**

In the revised manuscript we have made it clear that the smaller channel led to a more closed canopy that made the channel more challenging to navigate (see L123).

**L123 – Combine "To avoid view obstruction of the channel bed, the RPA was flown manually below the canopy." with earlier sentence about manual flights.**

In the revised manuscript we have removed this line and included the information in the earlier sentence.

**Fig 2 - a and b appear to show slightly different things - a appears to show oblique imagery collected along two flight lines, whereas in b the oblique imagery only seems to be collected along the flight lines closest to the bank.**

Figure 2 has been updated such that b now shows more detail with the oblique imagery collected along two flight lines.

**L137 - What form did the GCPs have?**

We have added additional detail to the revised manuscript highlighting that the GCPs were comprised of 10 x 10 cm ceramic tiles with a central X marking the surveyed location (see L140 of the revised manuscript).

**L138 - Were these actually targets that were placed in the field, or just points measured using the total station?**

We added additional text highlighting that all points were surveyed using the ceramic tiles as a target with the following in the revised manuscript:

"…with additional *tiles* positioned on the dry exposed bars and below the water surface in order to serve as independent checkpoints, to assess the accuracy of the model outputs."

**L143-146 - Move this to the start of the paragraph so that it is with the other material about what you did in the field.**

The line has been moved to the start of the paragraph.

**L150 - By reach, do you mean the channel segments, or the study sections?**

The intention was to refer to all the channel segments that we surveyed. This has been clarified in L154 of the revised text.

**L155 - I've not heard of this algorithm before - might be useful to explain briefly what the cloth resolution and max distance parameters refer to.**

Additional text has been added to L160-162 of the revised manuscript highlighting what these parameters are, as described by Zhang et al., 2016.

**L157 - How is this script implemented - in CloudCompare or something else?**

The script is implemented using a custom script developed by Dietrich for Python. This detail has been included in L164 of the revised text.

**L162 - Is 0.02 the resolution of the final point cloud? Make this clear if that is the case, or give the final resolution.**

The 0.02 m resolution refers to both the subsampled point cloud and DEM. This has been clarified in L169-170 of the revised text.

**L175 - How is 'mean roughness' defined? What size moving window was used?**

Mean roughness was defined as the average roughness value for each training site's roughness DEM. The roughness DEMs were generated in Cloud Compare, which assigns a roughness value to each point in the cloud. The technique used by Woodget and Austrums (2017) was then applied using a 1 m$^2$ window, which is similar to the size of the established training sites. These details have been included in the revised manuscript in L181-184.

**L180 – What is the fixed interval you used?**

We have reiterated in this line that the fixed interval we used was 1 m in L189.

**L185 - This suggests that you have separate data points for the water surface and the bed. Are the water surface points are produced when implementing Dietrich's routine?**

Yes, separate points for the water surface were extracted by generating water surface meshes across the channel and then running the Dietrich routine. This has been clarified in the revised manuscript in L195.

**L199- Not sure that this is quite the right phrase - more that you ran the PCA, and kept the first three components that it produced?**

The authors accept this comment. The line has been replaced with:

"The PCA was run and then three of the five components were retained for further analysis, which together explained approximately 79.0% of the variation in the dataset…"

**L210 - Is 1 m your spacing between observations? Does it matter than many of your properties are measured over a longer section of the channel (e.g. 15 m for slope)?**

Yes 1 m is the spacing between observations, which provides a smooth transition in morphologies across the channel (see L189 of the revised manuscript). While the slopes are a metric calculated over longer distances, they were still based on all observations in a window. For example, the 15 m slopes were based on a linear model fit through all 15 observations at 1 m increments in a window (see L196 of the revised manuscript).

**L221 - This might be clearer if you say that you first calculated the diversity metric for different sections of the channel, and then that you calculated the standard deviation of these values. The same applies to the first sentence after the table.**

This line, and the one following the table have been reorganized so calculation of the diversity metric is introduced first, and then the calculation of standard deviation.

**L235 - change to 0.11 m, and 0.025 m**

The change has been made in the revised manuscript.

**Figure 5 - I see that you've removed the sentence about in-stream wood from the paper following a reviewer comment, but given that the wood is visible in this figure, I'd move those sentences into this figure caption instead.**

Accepted, text has been added to the caption in Fig 5 describing how wood was digitized in the figure.

**L268 - I would use m or mm throughout, not cm. At least be contestant - the survey errors were a similar size to the grain sizes, yet you used m there and cm here.**

All instances of cm in the manuscript have been changed to m.

**Figure 9 - Clarify that these are the classifications produced by the clustering, not your field observations.**

**Specify flow direction.**

**Is the orthomosaic the images that the topo data was determined from? If so, how is there a hole in the image in b, but elevation data at that location in a? Might need to add something to the methods if interpolation was used in areas of sparse data.**

We edited Fig. 9 such that the flow direction is now specified in the revised manuscript. Regarding the hole in the orthomosaic, overhanging vegetation in this area led to a busy point cloud that led to the hole in the generated orthomosaic. However, this area was covered with the point cloud, and by using the Cloth Simulation Filter in CloudCompare, it was possible to generate the DEM of the bed in 9a. This is now detailed in the caption of Fig 9 in the revised manuscript.

**Table 3 title – Change to '...classification using clustering'?**

The title has been changed to "Accuracy assessment of morphological unit classification using k-means clustering."

**Line 351 - This took me a bit of thinking to understand. It might be helpful to point out something along the lines of: a lower standard deviation means that the diversity metric is more consistent between the different samples, which indicates that each individual sample is more likely to be a representative sample of the natural variability in the channel.**

Accepted, this suggested wording has been incorporated into the text of the revised manuscript in L364-367.

**LIST OF ALL CHANGES MADE TO MANUSCRIPT**

**-** Edits based on all comments from Associate Editor.

- L237, corrected typo: anymptote to asymptote.

- Updated references to the lower "3 km" of channel to "3.0 km" to be consistent.

- Updated the address of the authors.

- Acknowledgments: added text acknowledging funding received for the study.

[revised manuscript text omitted]